# Repurposing of promoters and enhancers during mammalian evolution

Francesco N. Carelli [1,3], Angélica Liechti[1], Jean Halbert[1], Maria Warnefors[2] & Henrik Kaessmann[2]

Promoters and enhancers—key controllers of gene expression—have long been distinguished from each other based on their function. However, recent work suggested that common architectural and functional features might have facilitated the conversion of one type of element into the other during evolution. Here, based on cross-mammalian analyses of epigenome and transcriptome data, we provide support for this hypothesis by detecting 445 regulatory elements with signatures of activity turnover (termed P/E elements). Most events represent transformations of putative ancestral enhancers into promoters, leading to the emergence of species-specific transcribed loci or 5′ exons. Distinct GC sequence compositions and stabilizing 5′ splicing (U1) regulatory motif patterns may have predisposed P/E elements to regulatory repurposing, and changes in the U1 and polyadenylation signal densities and distributions likely drove the evolutionary activity switches. Our work suggests that regulatory repurposing facilitated regulatory innovation and the origination of new genes and exons during evolution.

[1] Center for Integrative Genomics, University of Lausanne, CH-1015 Lausanne, Switzerland. [2] Center for Molecular Biology of Heidelberg University (ZMBH), DKFZ-ZMBH Alliance, D-69120 Heidelberg, Germany. [3] Present address: Wellcome Trust/Cancer Research UK Gurdon Institute, University of Cambridge, Cambridge CB2 1QN, United Kingdom. Correspondence and requests for materials should be addressed to F.N.C. (email: fr.carelli@gmail.com) or to H.K. (email: h.kaessmann@zmbh.uni-heidelberg.de)

Gene transcription in mammals is controlled by the interactions between proximal and distal gene regulatory elements. Promoters—the proximal regulatory regions associated with the transcription start site (TSS) of a gene—mediate the recruitment of the RNA polymerase II (Pol II) through their recognition by general transcription factors[1]. The spatiotemporal activation of gene expression is further defined by transcription factors bound to other regulatory loci, including TSS-distal enhancers[2,3]. Several sequence and structural features characterize promoters and enhancers. Most vertebrate promoters are CpG-rich[1], while most enhancers are CpG-poor[4], a difference that is also reflected in the respective regulatory motif compositions[5,6]. While both types of elements are characterized by accessible chromatin[7], enhancers and promoters have different chromatin modification profiles. Promoters are generally associated with higher levels of trimethylation of lysine 4 at histone 3 (H3K4me3) compared to monomethylation of the same residue (H3K4me1); whereas, the opposite pattern is found for enhancers in a poised state[8]. However, both types of elements are enriched for acetylation of lysine 27 at histone 3 (H3K27ac) when active[9,10].

Although the aforementioned features led to the distinction of promoters and enhancers as different types of regulatory elements, recent work unveiled similarities in their architecture and activity (reviewed in refs. [11–13]). Transcriptome analyses revealed that both promoters and enhancers are bidirectionally transcribed[4,14,15], and that this process involves the recruitment of the same transcriptional machinery[16]. Moreover, some regulatory elements display similar chromatin modification profiles despite different activities; for example, enrichment of H3K4me3 can also be detected at highly transcribed enhancers[17]. The two classes of regulatory elements may also show bivalent functionality, with some enhancers acting as alternative promoters[18] and some promoters enhancing the expression of other genes[19–21]. Although these observations blurred the boundary between the two classes of regulatory regions, the association of promoters to long transcripts that are 5′ capped and 3′ polyadenylated still distinguishes these regulatory elements from enhancers, which produce short, generally unstable transcripts[4].

Transcript stability has been linked to the relative enrichment of destabilizing polyadenylation signals (PAS) and stabilizing 5′ splicing (U1) motifs downstream of the TSS. U1 sites, apart from their role in splicing, prevent premature transcript cleavage from cryptic PAS through their binding with the U1 snRNP[22]. Polyadenylation signals proximal to the TSS have the opposite effect, and direct nascent transcripts towards exosome degradation[23]. Unidirectional promoters show an enrichment of U1 sites and a depletion of PAS sites in their sense direction relative to their upstream antisense direction, which supposedly limits pervasive genome transcription[24]. The instability of enhancer-associated transcripts is also due to an enrichment of PAS over U1 motifs[17].

Given the structural and functional similarities between enhancers and promoters, changes in the U1-PAS axis might in principle alter the activity of these regulatory elements. Inheritable mutations at PAS and U1 sites might stabilize enhancer-associated transcripts, thus facilitating their evolution into promoters[25]. Similarly, mutations might destabilize promoter transcription, but not affect the ability of these loci to regulate the expression of other genes. Thus, one might expect to observe orthologous regulatory elements that function as enhancers in one species but as promoters in another. Interestingly, recent work reported the frequent evolutionary emergence and decay of enhancers[26] and, at a lower rate, promoters[27] in mammals. Although, the gain- and loss-of-regulatory elements is largely driven by the insertion and deletion of genomic sequences, such as repetitive elements[26,28], many regions align to orthologous loci

in other species not showing the same functionality[26,27]. This raises the possibility that some regulatory elements might experience changes in their activity during evolution—a process we refer to as regulatory repurposing.

Suggestive evidence for the existence of regulatory repurposing events has been reported in mammals. We recently described an enrichment of enhancer-associated chromatin marks at mouse loci orthologous to the promoters of new rat-specific mRNA-derived gene duplicates (retrocopies)[29]. Moreover, two separate studies reported evidence of 11 mouse long-non-coding RNAs whose promoter sequences were orthologous to putative human enhancers[30,31]. Nonetheless, a thorough investigation of the prevalence of regulatory repurposing during mammalian evolution and its underlying molecular mechanisms has been lacking.

Here we report a detailed survey of regulatory repurposing in mammals. Based on integrated evolutionary analyses of mammalian chromatin profiles and transcriptional data, we detect 445 repurposed elements in sister species from the primate and rodent lineages. In most cases, putative ancestral enhancers were converted to promoters during evolution. This observation suggests that enhancers might have a higher repurposing potential than promoters. Enhancer-to-promoter transformations led to the origination of species-specific transcribed loci or 5′ exons. We also find that distinct GC sequence compositions and stabilizing 5′ splicing (U1) regulatory motif patterns may have predisposed P/E elements to regulatory repurposing, and that changes in the U1 and polyadenylation signal densities and distributions likely underlie the evolutionary activity alterations. Overall, our work highlights regulatory repurposing as a notable mechanism that likely facilitated regulatory innovation and the origination of new genes and exons during mammalian evolution.

## Results

**Regulatory element repurposing in primates and rodents**. As only limited evidence of putative regulatory repurposing was available from previous studies, we first sought to confirm its occurrence and study its prevalence in mammals. Toward this aim, we defined genome-wide sets of putative enhancers in a mammalian reference species and investigated whether any of these loci were orthologous to putative promoter regions from a closely related species (Fig. 1a) and hence represented candidate repurposed elements—here referred to as P/E elements. We focused our work on four species from two mammalian orders: human and rhesus macaque, as representatives of the primate lineage, and mouse and rat from the rodent lineage. We chose these two species pairs for several reasons: first, a large amount of gene expression and chromatin modification data is publicly available for human and mouse, allowing for the annotation of comprehensive sets of regulatory elements based on various tissues and developmental stages. Second, the relatively short evolutionary divergence times (25–29 millions of years) between human/mouse and macaque/rat, respectively, facilitates the definition of high confidence orthologous regions for each species pair, thus enabling the comparison of regulatory activities for large numbers of genomic loci. Third, suitable outgroup species (marmoset for the two primates and rabbit for the rodents) with relevant data are available for evolutionary inferences. Finally, both species sets have respective advantages and disadvantages, and therefore the analyses of both datasets allow for overall optimal analyses. For example, while the low mutation rate and resulting high sequence similarity in the primates may allow for higher confidence inferences of early regulatory element evolution, the larger rodent sequence divergence and more efficient natural selection during rodent evolution may

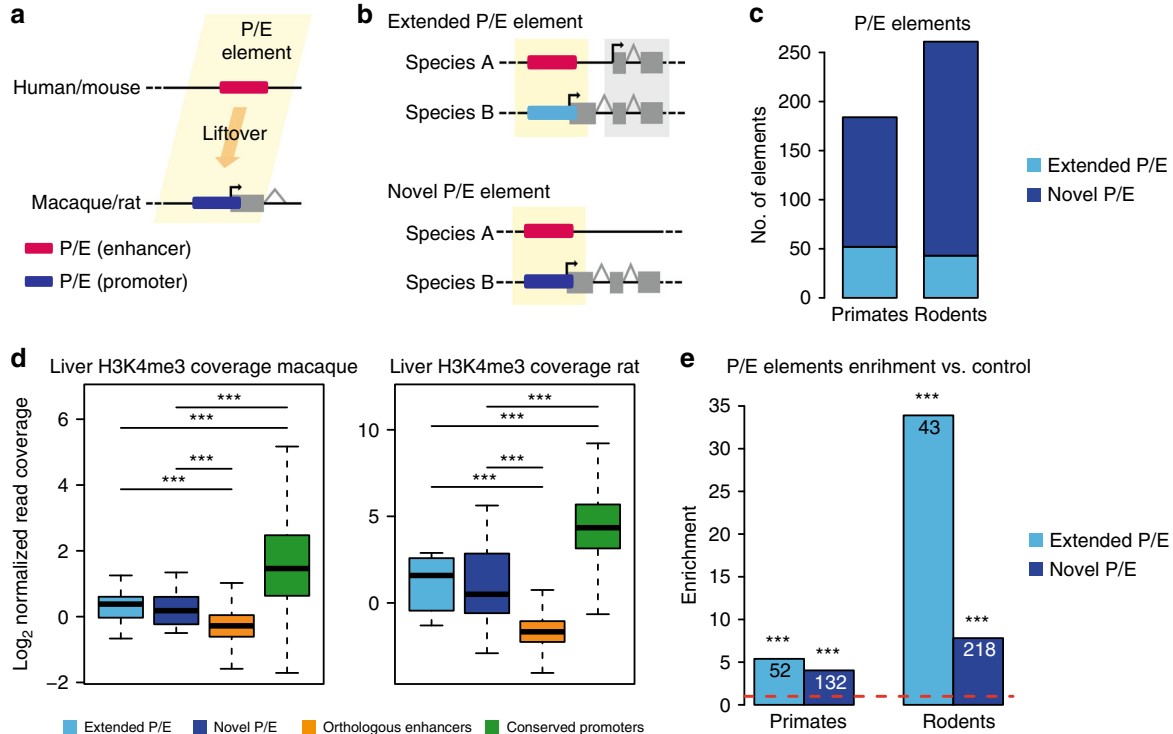

**Fig. 1** Repurposing of regulatory elements in mammals. **a** Schematic representation of a P/E element in primates and rodents. **b** Types of P/E elements. **c** Total number of novel and extended P/E elements detected in primates and rodents. **d** Macaque and rat H3K4me3 ChIP-seq reads density from liver (log2 read count normalized by input read count) measured at novel and extended liver P/E elements, loci orthologous to the sister species liver enhancers and not associated to any stable TSS, and stable liver promoters conserved in the sister species. Whiskers up to 1.5 times the interquartile range; outliers removed for graphical purposes. Significant differences (Mann–Whitney $U$-test with Benjamini-Hochberg correction): ***$P$ < 0.001. **e** Fold-difference between P/E elements ratio (fraction of human or mouse enhancers corresponding to promoters in sister species) and P/inactive ratio (fraction of human or mouse inactive regions corresponding to promoters). The red line indicates no difference between the two ratios. Numbers indicate the number of P/E elements for each group. Significant differences (Fisher's exact test with Benjamini-Hochberg correction): ***$P$ < 0.001

allow for easier and/or more powerful detection of regulatory repurposing events.

We defined sets of putative promoters as the upstream regions of stably transcribed loci, assembled using both our newly generated as well as recently published[32] strand-specific RNA-seq data from four organs (brain, heart, kidney, and liver; see Methods) (Supplementary Data 1–2), which yielded between 27,042 and 33,520 promoters for each species (Supplementary Data 3). We then identified putative enhancers in human and mouse (i.e., our reference species for which extensive relevant data are available—see above) by combining transcription, DNase hypersensitivity and histone modification data from the same set of organs. Specifically, we extracted all DNase hypersensitive sites enriched for H3K4me1 and/or H3K27ac in any of the four organs, and filtered out those regions which showed signatures of promoter activity in any sample from a broad set of tissues and cell types (see Methods). This was done to limit the inclusion of potential bivalent elements (i.e., elements characterized by both enhancer and promoter activity in different tissues of the same organism), which would hamper our search for bona fide repurposed elements. Nonetheless, we cannot exclude that some putative enhancers might have promoter activity in other tissues or developmental stages. Additionally, we included a second set of putative enhancers, defined using CAGE data from a number of organs and cell lines in human and mouse[4], which was further filtered for bivalent elements. Overall, we obtained a high-confidence set of 110,611 putative enhancers (hereafter simply termed enhancers) in human and 127,145 in mouse (Supplementary Data 4).

**Functional remodeling facilitated regulatory innovation.** After the annotation of putative regulatory regions in our set of species, we assessed the evolutionary conservation of their activity to detect P/E elements. Specifically, we extracted 97,451 human and 105,700 mouse enhancers that could be aligned to orthologous sequences in their sister species. We then investigated whether the orthologous loci in the macaque or rat genome, respectively, overlapped the promoter region of a stable transcript (Fig. 1a, Supplementary Data 5). After removal of putative false positives associated to short transcript isoforms (46 in primates, 60 in rodents, see Methods), we thus identified 184 P/E elements in primates and 261 elements in rodents (i.e., 445 in total). We subdivided the P/E elements into two distinct categories based on their association to a species-specific (i.e., macaque- or rat-specific) transcribed locus (novel P/E) or to a new species-specific 5′ exon of a locus transcribed in both species of the respective lineage pair (extended P/E) (Fig. 1b, c, Supplementary Data 6). To further confirm the promoter activity of P/E elements in macaque and rat, we inspected their chromatin state using publicly available H3K4me3 ChIP-seq data from adult liver samples[26]. Consistent with the notion of regulatory repurposing, P/E elements associated with stable liver transcripts in macaque and rat display higher H3K4me3 coverage compared to sequences orthologous to non-repurposed liver enhancers in human or mouse, respectively (Fig. 1d). H3K4me3 levels at novel and extended P/E elements in macaque and rat are significantly lower compared to those measured at stable liver promoters conserved in their sister species (Fig. 1d), likely reflecting the lower expression level of P/E-associated transcripts (Supplementary Fig. 1). Our analysis

| | Outgroup species | Converted liver P/E elements | Ancestral liver P/E promoters | Ancestral liver P/E enhancers |
|---|---|---|---|---|
| **Table 1 Directionality of mammalian repurposing events in liver** | | | | |
| Primates | Marmoset | 66 | 1 (1.5%) | 27 (40.9%) |
| Rodents | Rabbit | 49 | 1 (2%) | 17 (34.6%) |

thus uncovered the presence of hundreds of P/E elements showing divergent regulatory activities in two major mammalian lineages. Notably, a sequence conservation analysis reveals that the sets of P/E elements overall show signatures of selective preservation (Supplementary Fig. 2), which suggests that at least subsets of them have acted as functional regulators at least during some time of their evolutionary history.

The detection of P/E elements in two closely related species could in principle result from the independent de novo evolution of distinct activities from an ancestral inactive region, rather than from a species-specific repurposing of an ancestral regulatory region. We therefore investigated whether the enhancer signature detected at a specific locus would significantly increase the chance of observing promoter activity at the orthologous locus in the closely related species, which would support the repurposing scenario. We defined control regions showing no signature of regulatory activity and no overlap with any exonic sequence in human and mouse (Methods section) and then evaluated whether their orthologous regions in the sister species were associated to the TSS of a stable transcript. Only 0.04% (79/181,688) of the control regions tested in primates and 0.02% (23/83,288) of those in rodents show this behavior. These percentages are significantly lower compared to the fraction of P/E elements retrieved in primates (0.18%, 184/97,405 elements; >4.03-fold enrichment for novel or extended P/E loci, Fisher's exact test, $P < 10^{-9}$) and rodents (0.24%, 261/105,640 elements; >7.81-fold enrichment, Fisher's exact test, $P < 10^{-8}$) (Fig. 1e). Although there is a difference in GC content between the inactive regions and the putative enhancers tested in both species (Supplementary Fig. 3), rat- and macaque-specific promoters are nonetheless more often orthologous to enhancers than to inactive loci with matched sequence composition in their sister species (Supplementary Fig. 4). These data corroborate the hypothesis that P/E elements likely correspond to ancestral regulatory regions that experienced evolutionary changes in their regulatory activity in the last 25–29 millions of years. Ancestral regulatory capacities of genomic sequences therefore facilitated regulatory innovation in mammals, while our analyses also suggest that a sizeable number of lineage-specific regulatory elements may have emerged de novo from the large inactive portion of the genome.

**Enhancers are the main source of regulatory repurposing.** The retrieval of hundreds of lineage-specific P/E elements allowed us to investigate at a broad scale the directionality of regulatory activity changes; that is, to define whether an ancestral enhancer evolved into a promoter, or vice versa. We thus investigated the presence of regulatory activity associated to regions orthologous to P/E elements in an outgroup species, in order to infer their ancestral state. Using ChIP-seq and transcription data to annotate putative regulatory elements in adult marmoset liver[26], we find that 40.9% (27 of 66) primate P/E elements with activity in liver and aligned to the marmoset genome correspond to orthologous putative enhancers in this outgroup species, whereas only 1.5% overlap a promoter (Table 1). Similarly, 34.7% of rodent P/E elements correspond to putative enhancers in rabbit, while only 2% overlap a promoter (Fig. 2a, Table 1). The higher fraction of ancestral P/E elements with enhancer activity suggests that most

repurposed elements correspond to ancestral enhancers that recently evolved species-specific promoter activities.

Our analysis also revealed that 38 (57.6%) and 31 (63.3%) P/E elements active in liver in primates and rodents, respectively, do not bear any signature of liver activity in the outgroup species. We therefore reasoned that some P/E elements might be active in a different organ in the outgroup. To evaluate this possibility, we determined the fraction of primate and rodent P/E elements orthologous to a promoter in any of the adult organs investigated in marmoset and rabbit. This analysis shows that only 8 of the 174 marmoset regions orthologous to a primate P/E element (4.5%) overlap the TSS of a transcript and therefore likely correspond to an ancestral promoter. A similarly low fraction (5/134, 3.7%) of rodent P/E elements correspond to promoters in rabbit. These results further confirm that only a limited number of detected repurposing events involved ancestral promoter elements.

Recent work reported a higher evolutionary turnover of enhancers compared to promoters in mammals[26]. To evaluate whether this bias could explain the observed higher enhancer-to-promoter repurposing rate, we compared the rates of repurposing and loss of activity for ancestral liver enhancers and promoters in both lineages (Fig. 2b; Supplementary Table 1; Methods). The rate of enhancer loss (≈50.5%, 2655 of 5260 ancestral enhancers) is ~1.92 times higher than that of promoter loss (≈26.3%, 1342 of 5110 ancestral promoters), in agreement with previous observations[26]. Notably, however, the rate of ancestral enhancer repurposing (≈0.51%, 27/5260) was ~26 times higher than that of ancestral promoter repurposing (≈0.02%, 1/5110). This pattern corresponds to an ~13-fold enrichment of enhancer repurposing relative to promoter repurposing (Fisher's exact test, $P < 10^{-3}$; Table 2), taking the increased enhancer loss rate into account. The lack of a statistically significant similar pattern in rodents (≈6-fold enrichment, Fisher's exact test, $P = 0.056$; Table 2) is likely due to lack of power and explained by evolutionary differences between the primate and glires (the clade including rodents and lagomorphs) species investigated. That is, the considerably larger divergence time of the glires species (divergence time: ≈80 million years between mouse/rat and rabbit) compared to that in primates (≈42 million years between human/macaque and marmoset), their higher mutation rates, and the more efficient natural selection in glires may obscure actual rates of activity turnover in this lineage. In any event, our primate analyses suggest that the higher rate of enhancer repurposing cannot be explained by the higher turnover rate of enhancers compared to promoters, although future work on additional closely related sets of species and organs will be needed to confirm this pattern.

**P/E elements have distinct sequence compositions.** Large-scale surveys have demonstrated that the motif and sequence composition of mammalian enhancers closely resembles that of promoter regions that do not overlap with CpG islands (CGIs), but differences between distinct types of enhancers have been reported[4]. Due to the peculiar evolutionary change in activity of P/E elements, we therefore investigated whether their sequence would be distinct relative to other regulatory regions. P/E enhancers in both human and mouse have an overall lower GC

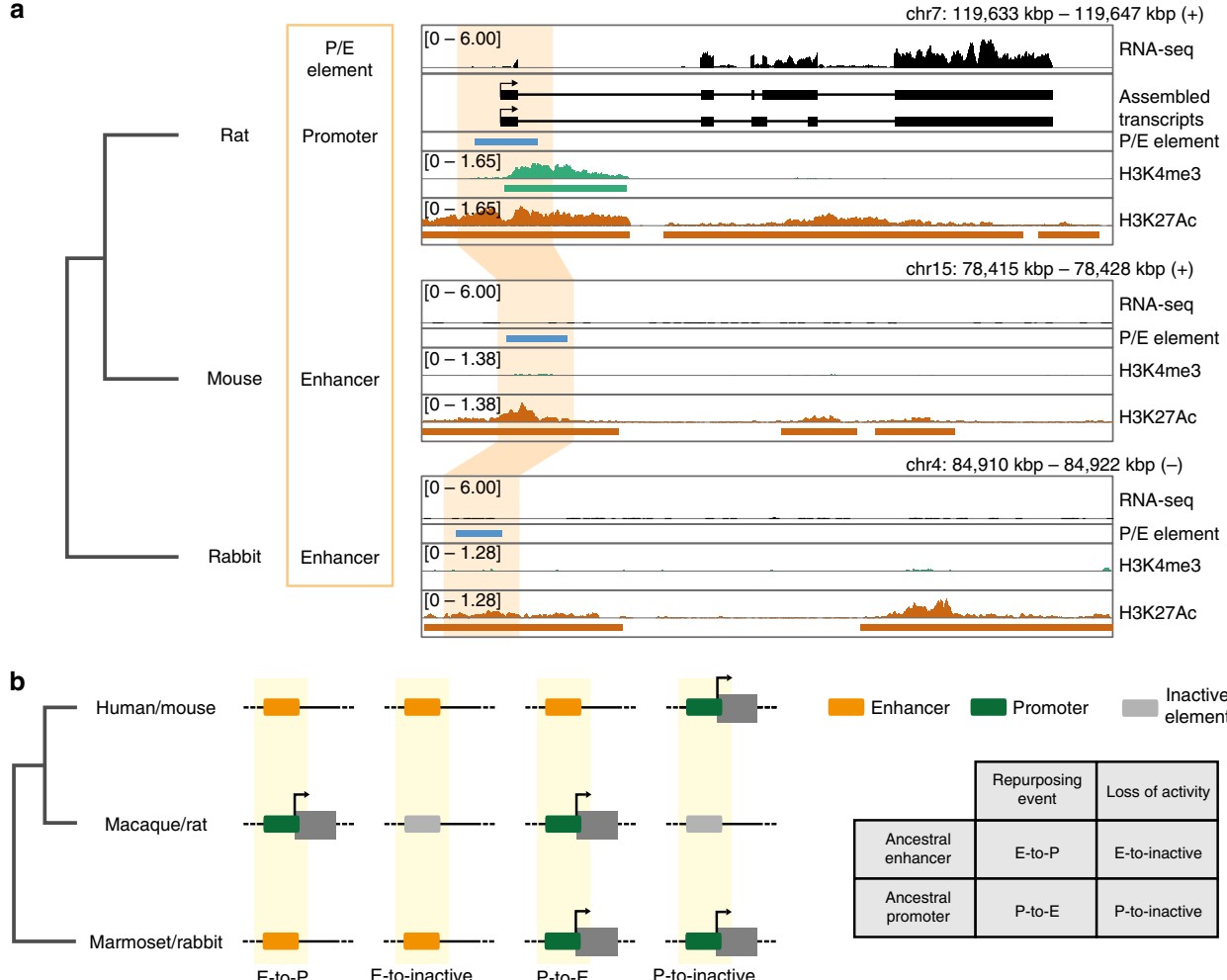

**Fig. 2** Repurposing of an ancestral glires enhancer. **a** The coordinates of a rodent P/E element, with enhancers activity in mouse and promoter activity in rat, are projected onto the rabbit genome (orange shaded area). The syntenic region in rabbit overlaps an H3K27ac peak but no H3K4me3 peak. The presence of a transcript in rat and absence of transcripts in mouse and rabbit are evident based on the RNA-seq tracks. For each species (from top to bottom) are shown: the RNA-seq coverage from liver; the assembled transcripts (only in rat); the P/E element locus; the liver H3K4me3 and H3K27Ac coverage and peaks from Villar et al. (2015). **b** Scheme depicting the different types of turnover events (Regulatory repurposing vs evolutionary loss) for ancestral enhancers and promoters

### Table 2 Turnover of regulatory elements activity in primates and glires

|  |  | Repurposing event | Loss of activity |
|---|---|---|---|
| Primates | Ancestral enhancer | 27 | 2655 |
|  | Ancestral promoter | 1 | 1342 |
| Glires | Ancestral enhancer | 17 | 1157 |
|  | Ancestral promoter | 1 | 417 |

and CpG content when compared with CGI-associated promoters (Fig. 3a–d), in agreement with known differences between CpG islands and enhancers[4]. Surprisingly, P/E enhancers have a significantly higher GC content compared to other enhancers and, to a lower extent, non-CGI-associated promoters, indicating that the sequence composition distinguishes P/E enhancers from other regulatory sequences (Fig. 3a, b). Similarly, there is a higher content of the CpG dinucleotide in P/E enhancers compared to other enhancers but not to non-CGI promoters (Fig. 3c, d), reinforcing the distinction of this class of regulatory elements from other enhancer elements. We also find that P/E promoters

in macaque and rat have significantly higher GC and CpG content than macaque/rat sequences orthologous to enhancers in human/mouse (Supplementary Fig. 5), although, we note that a large number of these orthologous loci might be inactive in macaque and rat. Small or no differences in GC and CpG content were instead evident between P/E elements and non-CGI promoters in macaque and rat (Supplementary Fig. 5). The overall differences in sequence composition between P/E elements and other regulatory regions are also reflected in their relative enrichment for core promoter-associated motifs with different GC contents (Supplementary Data 7). Taken together, these results highlight how the sequence and motif composition of P/E elements distinguishes them from other regulatory elements.

**Sequence compositional changes associated with repurposing.** Next, we sought to trace whether compositional changes may underlie functional shifts of P/E elements. Notably, the CpG content of regulatory elements has been proposed to influence their transcriptional output[33,34]. CpG islands are usually associated to the promoter of broadly and highly expressed genes[1], where they favor gene expression by creating a nucleosome-free environment[33,34]. We therefore asked whether P/E elements with

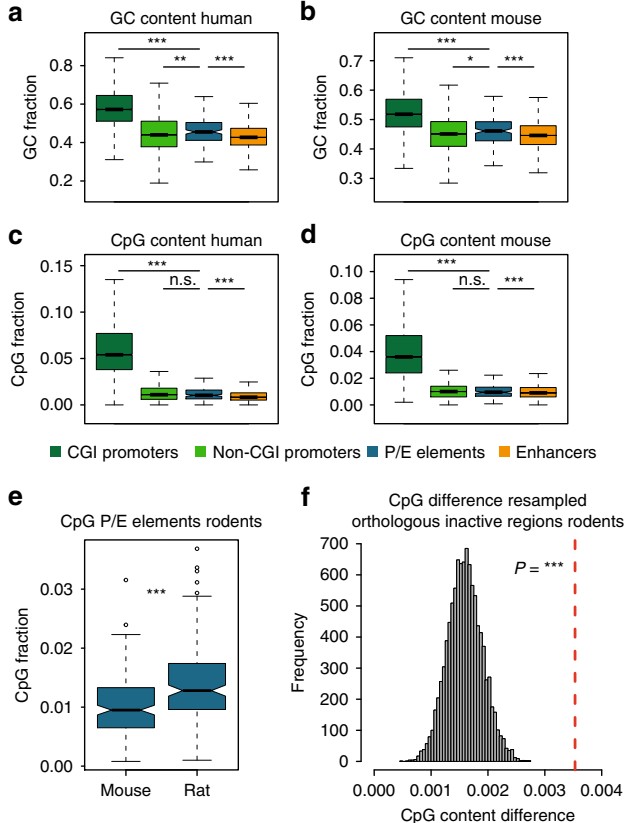

**Fig. 3** Nucleotide composition of P/E elements. **a–d** Distribution of GC and CpG content for different classes of regulatory elements in human and mouse. Whiskers up to 1.5 times the interquartile range; outliers removed for graphical purposes. Significant differences (Mann–Whitney $U$-test with Benjamini-Hochberg correction): ***$P < 0.001$; **$P < 0.01$; *$P < 0.05$; (n.s.) $P \geq 0.05$. **e** Distribution of CpG dinucleotide frequency in orthologous rodent P/E elements. Whiskers up to 1.5 times the interquartile range. **f** CpG frequency difference measured between orthologous inactive regions. The distribution was obtained by resampling 10,000 times a number of orthologous inactive regions equal to the number of P/E elements in **d** and calculating the average CpG frequency difference. The red line indicates the mean CpG frequency difference between orthologous rodent P/E elements. The $P$-value indicates the fraction of resampled inactive regions with an average CpG frequency higher than that of P/E elements ***$P < 0.001$)

promoter activity were associated to higher GC and CpG contents compared to their orthologous regions. Importantly, the genome-wide sequence composition—and particularly the GC content—can vary even between closely related species[35]. In this case, any difference observed between orthologous P/E elements should be compared to the global change in sequence composition experienced by the two species, using regions not subject to natural selection. To do so, we compared the difference in GC and CpG frequencies between orthologous P/E elements with that observed between orthologous inactive regions. We found that rat P/E promoters showed somewhat higher GC content compared to their orthologous mouse P/E enhancers but that this difference was not significantly stronger than that of the control regions (Supplementary Fig. 6), in agreement with the previously reported overall higher genome-wide GC-content in rat[35]. By contrast, we noted that the total content of CpG dinucleotides in rat P/E promoters is significantly increased compared to the control regions (Fig. 3e, f), indicating that the activity turnover of P/E loci is mirrored by a change of CpG frequency in this lineage.

In primates, the GC content did not differ significantly between the orthologous P/E element sequences; whereas, the frequency of CpG dinucleotides was significantly higher in macaque P/E promoters compared to the human P/E enhancers (Supplementary Fig. 7). Notably, the enrichment in both GC and CpG content measured in macaque was statistically significant when compared to the control regions (Supplementary Fig. 7). The observed GC enrichment reflects the slightly lower GC content of the macaque genome compared to the human one[35]. Overall, the higher CpG content in P/E promoters together with the reported effect of CpG content on transcription[33,34] suggest that specific changes in nucleotide composition contributed to the regulatory repurposing of P/E elements both in rodents and in primates.

**Distinct U1 motif and PAS patterns at P/E elements**. While promoters and enhancers both have the inherent capacity to promote transcription, only promoters generate stable transcripts[17]. In mammals, this difference between the two types of elements seems to be mainly directed by specific signals downstream of the TSS. U1 sites are commonly enriched downstream of promoters and depleted in the antisense orientation as well as around enhancers, whereas PAS generally follow the opposite trend[4,24]. Owing to their potential role in transcription, we compared the distribution of U1 signals and PAS around orthologous novel P/E elements (Fig. 1b, lower panel). For each element, we extracted U1 and PAS motifs up- and downstream of the TSS of their associated transcript in macaque and rat, as well as for the corresponding orthologous regions in the respective sister species (Fig. 4a). In both macaque and rat, as expected given the promoter activity of the P/E elements, the density of U1 sites downstream of the TSS is higher compared to the antisense orientation, whereas there is a weak but significant opposite trend for PAS motifs (Fig. 4b, Supplementary Fig. 8). In human and mouse, consistent with the lack of stable transcripts associated to P/E elements, there are weak to no differences in U1 or PAS distribution around each P/E element (Fig. 4c, Supplementary Fig. 8). We further compared the U1/PAS density in human and mouse at P/E enhancers to that measured around the TSS of CAGE-defined non-P/E enhancers elements, similarly characterized by the lack of stable transcription. Contrary to what is observed for these non-P/E enhancers, PAS density is significantly lower on either side of P/E enhancers, and U1 density is higher downstream but not upstream of the projected TSS (Supplementary Fig. 8). In light of the higher enhancer-to-promoter repurposing rate (see above), these results suggest that the unique sequence and motif composition distinguishing P/E enhancers from other typical enhancers may predispose them to repurposing into novel promoters during evolution.

**Evolutionary changes in the U1/PAS axis**. Evolutionary changes in the U1/PAS axis have been proposed as a mechanism underlying the emergence of new transcribed loci that may be selectively preserved and thus form new genes[25]. However, so far, evidence supporting this hypothesis has been limited. We therefore took advantage of our dataset of orthologous P/E element pairs to test whether evolutionary changes in the U1 and PAS motif distributions around these loci might underlie their regulatory activity transformation. A comparison of the distribution of U1 sites surrounding orthologous P/E elements in rodents reveals that their density is significantly higher over 1 kilobase (kb) downstream of the TSS of rat P/E promoters than in the orthologous non-transcribed regions of mouse P/E enhancers (mean of 2.83 vs. 2.25 U1 sites per kb, Wilcoxon's test, Benjamini-Hochberg corrected $P < 10^{-4}$, Fig. 4d), whereas there

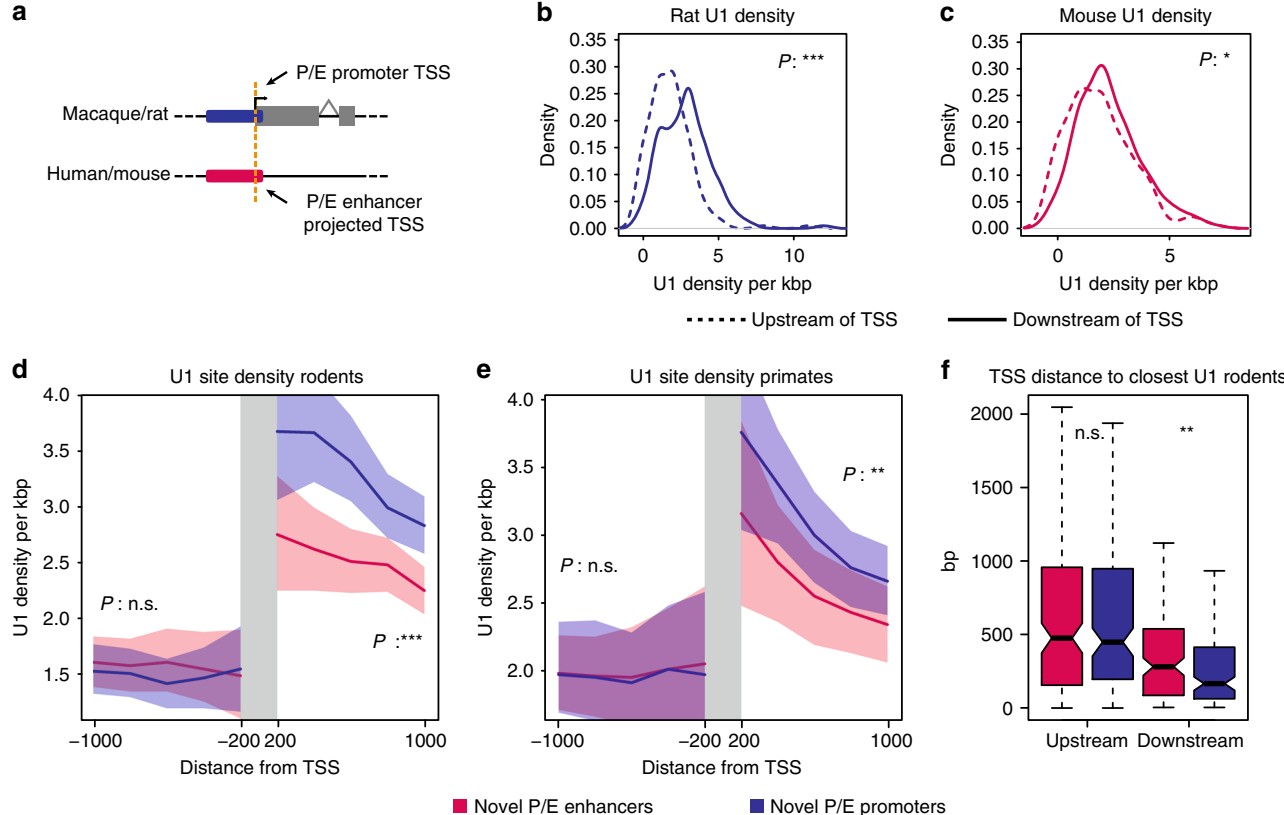

**Fig. 4** U1 site distribution at P/E elements. **a** Schematic representation of an orthologous P/E element. The orange line depicts the position of the TSS in rat and of the projected location in mouse. **b**, **c** Distribution of U1 site density per kb upstream (dashed line) and downstream (continuous line) of the TSS of transcripts associated to novel P/E elements in rat **b** and mouse **c**. Significant differences (Mann–Whitney $U$-test with Benjamini-Hochberg correction): ***$P < 0.001$; *$P < 0.05$. **d**, **e** Cumulative density of U1 sites up- and downstream of novel P/E-associated TSSs in rodents and primates, respectively. Lines represent the mean U1 density (per kb) over 200, 400, 600, 800, and 1000 nt-long windows from the TSS, shaded areas represent 95% confidence intervals. Significant differences (Mann–Whitney $U$-test with Benjamini-Hochberg correction): ***$P < 0.001$; **$P < 0.01$; (n.s.) $P \geq 0.05$. **f** Distribution of up- and downstream distances of the closest U1 site from each novel P/E-associated TSS in rodents. Whiskers up to 1.5 times the interquartile range; outliers removed for graphical purposes. Significant differences (one-tailed Mann–Whitney $U$-test with Benjamini-Hochberg correction): **$P < 0.01$; (n.s.) $P \geq 0.05$

is no significant difference in the corresponding upstream regions. Analyses of PAS distributions for the same regions only reveal a weak decrease in PAS density downstream of the TSS of rat P/E promoters (mean of 1.45 vs. 1.70 PAS sites per kb, Wilcoxon's test, Benjamini-Hochberg corrected $P < 10^{-2}$, Supplementary Fig. 9) and no difference in the antisense orientation. In primates, we find a significant trend in U1 motif distribution around orthologous P/E elements similar to that detected in rodents (Fig. 4e) but no difference in PAS density (Supplementary Fig. 9), probably due to the low sequence divergence between human and macaque and the resulting lack of power[36]. Furthermore, the distance separating the TSS from the closest downstream U1 site in P/E promoters is significantly shorter than that for orthologous P/E enhancers (one-tailed Wilcoxon's test, Benjamini-Hochberg corrected $P < 10^{-2}$; Fig. 4f, Supplementary Fig. 10). Finally, U1 sites preceded a PAS downstream of rat P/E promoter TSSs in rat in 84.8% of the cases, compared to 66.8% for the orthologous mouse P/E enhancers (Fisher's exact test, $P < 10^{-3}$), with no significant differences in primates. Altogether, our analyses reveal evolutionary shifts in the distribution of U1 sites and, to a lesser extent, PAS motifs, which mirror the presence or absence of stable transcripts (i.e., promoter or enhancer activity) at P/E loci. Thus, changes in the U1/PAS axis may contribute to the origination of promoters from enhancers and, as a consequence, the emergence of new transcribed loci and, ultimately, functional new genes in mammals.

## Discussion

Mammalian promoters and enhancers share many similarities in their chromatin architecture, and—apart from a minor fraction of bivalent elements[20]—these regulatory loci are best distinguished based on the stability of their associated transcripts[17]. This suggests that small changes in the DNA sequences underlying or surrounding regulatory regions could redefine their activity. In our work, we provide support for this hypothesis by identifying hundreds of mammalian elements that experienced an evolutionary turnover in their regulatory activity, and by tracing specific sequence changes that accompanied this process.

Previous attempts to identify evolutionarily repurposed regulatory sequences uncovered 11 mouse lncRNA promoters that were orthologous to putative enhancer elements in human[30,31]. The low number of identified candidate elements is likely due to the long evolutionary distance separating the two species. Notably, the divergent activity in these cases might not necessarily result from a repurposing event, but could rather be the result of independent evolution. To overcome these issues, we focused on the comparison of more closely related species in our study, and we used inactive genomic regions as controls to assess whether the divergent activity of P/E elements likely results from evolutionary switches in their function. The hundreds of P/E elements uncovered here indicate that the evolutionary turnover of regulatory element activity is much more extensive than could have been estimated based on the human/mouse comparisons, but

even so, we have likely underestimated the number of functionally repurposed elements. For example, we conservatively excluded a large number of putative bivalent regulatory elements (characterized by enhancer and promoter activity in the same species) in order to maximize the confidence in detecting true turnover events, which may have led to the removal of many real enhancers characterized by H3K4me3 enrichment[17]. Through the integration of chromatin profiles and transcriptome data from additional organs and from multiple species, future analyses will allow to more precisely estimate the occurrence of regulatory repurposing in mammals. In any event, our work indicates that the repurposing of regulatory elements activity is a widespread process shaping the mammalian regulatory landscape.

We note that our ability to detect cases of a functional enhancer repurposed into a functional promoter (and vice versa) is highly dependent on the definition of functional regulatory regions. In this regard, it is important to remember that enhancers defined through their enrichment for specific histone modifications tend to have relatively low in vitro validation rates (~50%)[37], although recent work showed that the number of validated elements may be underestimated by episomal-based essays[38]. It is possible, therefore, that the changes in transcriptional stability produced by some repurposing events might not involve functional enhancers in human and mouse. However, the putative enhancer activity of around 32% and 17% of all P/E elements in human and mouse, respectively, was predicted based on their unstable bidirectional transcription, which has a higher validation rate compared to other essays[4] (67.4–73.9%). Notably, when considering only CAGE-defined enhancers in our dataset, we still observe a significantly higher fraction of repurposing events compared to those detected in our control set of inactive regions (0.18%/0.28% vs. 0.04%/0.02% in primates/rodents, Fisher's exact test $P < 10^{-14}$), supporting the hypothesis that repurposing events do involve *bona fide* enhancers that evolve into promoters able to drive stable transcription in mammals.

Our investigation of P/E element activity in outgroup species suggests that most turnover events seem to involve the repurposing of ancestral enhancer elements into species-specific promoters, and that this observation cannot be solely explained by the higher evolutionary turnover of enhancers compared to promoters in mammals[26]. It should be noted that more than half of the alignable P/E elements in each lineage had no detectable activity in the outgroup species. This is likely due to the relatively large evolutionary distance that separates our core set of species from their evolutionary outgroups. Consequently, the regulatory activity of these loci might either have emerged after the split of the outgroup lineages or might have been lost during the evolution of the outgroup species lineages. Although we cannot exclude that the inferred directionality of the repurposing process is influenced by the lack of definition of the ancestral state for part of the P/E loci, such a scenario is unlikely to fully explain the biased enhancer-to-promoter conversion pattern. On the contrary, the higher rate of enhancer turnover should in principle disfavor the detection of enhancer-to-promoter turnover events, given that it reduces the likelihood of detecting enhancers conserved in more distantly related species. Our results therefore suggest the existence of differences in repurposing potential between enhancers and promoters, which could involve their underlying DNA sequence and/or aspects of their chromatin composition. Future work, involving more closely related species, or different populations of the same species, will be necessary to further explore the biased directionality of the repurposing process and uncover its mechanistic bases.

The sequence analysis of P/E elements revealed features that distinguish these loci from other regulatory loci, and it provided initial evidence for the potential mechanisms behind the repurposing process. Notably, the high GC and CpG content could make P/E loci particularly prone to drive the expression of neighboring sequences, for example through the recruitment of CpG-binding proteins, such as Cfp1[39]. This protein is known to deposit H3K4me3 marks over the bound sequence[40], which in turn seems to favor transcription through different mechanisms[41,42]. Moreover, recent work showed how CpG sites favor promoter over enhancer activity in a massively parallel regulatory element assay in mouse[43]. The significantly higher CpG content of P/E elements with promoter activity strongly suggests that the fixation of nucleotide substitutions contributed to the turnover events by increasing (or decreasing) the density of this dinucleotide, leading to the creation or disruption of specific motifs that altered transcriptional capacities.

Moreover, U1 site density shifts also seem to be involved in the repurposing process. A higher number and a higher proximity of U1 sites characterize the region downstream of the TSS of P/E-associated transcripts, compared to their transcriptionally inactive orthologous regions. U1 sites are thought to promote transcript stability in mammals, suggesting that changes in the distribution of these motifs might be responsible for the stabilization or destabilization of P/E-associated RNAs. On the other hand, it is unclear whether the redistribution of polyadenylation signals (PASs) has had a significant influence on the turnover processes. Although PASs are slightly depleted downstream of the TSS compared to the upstream region, we found little to no differences in PAS distribution between orthologous P/E elements. Therefore, at least in this context, variation in U1 site distribution could be sufficient to drive the repurposing process.

The finding of numerous P/E elements raises the question of what impact repurposing events have on mammalian phenotypic evolution. We believe the influence of this process regarding the (ancestral) enhancer function of P/E elements is likely limited. The enhancer activity exerted by many mammalian promoters suggests that the emergence of stable transcripts associated to P/E elements might not abolish their distal regulatory activity[20]. Additionally, the potential loss of enhancer activity of a P/E element upon its regulatory repurposing might not affect the expression of its target gene(s), as it could be compensated for by the presence of other enhancers[44,45]. On the other hand, repurposing events might be phenotypically relevant through their contribution to the process of the birth and death of genes and transcripts. Our results provide solid evidence for the emergence of lineage-specific, stable transcripts from former enhancer elements. These transcripts may, potentially, represent new genes, which would support the hypothesis put forward by Wu and Sharp[25]. Overall, our analysis highlights regulatory repurposing as a mechanism underlying molecular innovation in mammals and calls for future work to unveil the impact of the repurposing process on mammalian phenotypic evolution.

## Methods

**RNA-seq data production and processing**. We generated 78 single-end strand-specific RNA-seq libraries from brain, heart, kidney, and liver samples for six mammals (human, macaque, marmoset, mouse, rat, and rabbit; Supplementary Table 2). Human samples derive from both adult and postnatal developmental stages, while all other mammalian samples stem from adult individuals (Supplementary Table 2). RNA was extracted from each sample using the RNeasy protocol from QIAGEN. RNeasy Micro columns were used to extract RNA from small (<5 mg) or fibrous samples and RNeasy Mini columns were used to extract RNA from larger samples. The tissues were homogenized in RLT buffer supplemented with 40 mM dithiothreitol (DTT) or QIAzol. RNA quality was assessed using the Fragment Analyzer (Advanced Analytical). The RNA-seq libraries were created using the TruSeq Stranded mRNA LT Sample Prep Kit (Illumina). Libraries were sequenced using the Illumina HiSeq 2500 to produce 100 nucleotide (nt) reads. The resulting transcriptome data were combined with a set of recently published transcriptome data[32]. Our study complies with all relevant ethical regulations with respect to both human samples and samples for the other mammals. Human samples were obtained from official scientific tissue banks or dedicated companies;

informed consent was obtained by these sources from donors prior to death or from next-of-kin. The use of all human samples for the type of work described in this study was approved by an Ethics Screening panel from the European Research Council (ERC) (associated with H.K.'s ERC Consolidator Grant 615253, Onto-TransEvol) and local ethics committees; that is, from the Cantonal Ethics Commission Lausanne (authorization 504/12) and Ethics Commission from the Medical Faculty of Heidelberg University (authorization S-220/2017). The use of all other mammalian samples was approved by an ERC Ethics Screening panel (ERC Consolidator Grant 615253, OntoTransEvol).

RNA-seq reads were aligned to the assembled genomes of their corresponding species (all genomes obtained from the UCSC website; human: hg38; macaque: rheMac8; marmoset: calJac3; mouse: mm10; rat: rn6; rabbit: oryCun2) using STAR[46] (version 2.2.1) using the 2-pass mapping mode and standard settings. Aligned reads from all replicates of each organ, totaling on average >100 million mapped reads, were used to reconstruct transcripts through a genome-guided de novo transcriptome assembly using StringTie[47] (version 1.3.4d) with the following parameters: -j 5 –g 50. Assembled transcripts from each organ were then merged using Cuffmerge[48] to define a unique set of transcripts. Expression levels (measured in FPKM) of the assembled transcripts were calculated with Cuffnorm[48] (version 2.2.1); we considered as stable all transcripts with a mean FPKM > 1 across replicates from the same organ and length (introns included) >1000 nt.

**ChIP-seq and DNase-seq data processing.** The chromatin data used in our studies derive from different sources. DNase, H3K4me3, H3K4me1, and H3K27ac data for mouse brain, heart, kidney, and liver (core dataset) were obtained from the Mouse ENCODE database[49] (Supplementary Data 8). DNase, H3K4me3, H3K4me1, and H3K27ac data from human brain, heart, kidney, and liver were obtained from the ENCODE database[50] or from the human Epigenome Roadmap database[51] (Supplementary Data 8). For both species, we also downloaded H3K4me3 data from additional adult and developmental samples from the same databases (extended dataset) (Supplementary Data 8). All processed data corresponding to an older genome assembly version were converted to the newest assembly version using LiftOver[52]. As data were processed in different ways, we applied a common approach to have comparable datasets. Specifically, we downloaded processed peaks (in narrowPeak format) from multiple replicates of all samples, and subsampled the top 20′000 (for H3K4me3 data) or top 80′000 peaks (for H3K4me1 and H3K27ac), ranked based on their peak score; all DNase hypersensitive site (DHS) peaks from each sample were retained as their numbers did not differ significantly across samples. We created organ/tissue-specific sets of H3K4me3, H3K4me1, and H3K27ac peaks by considering loci shared by at least three replicates from each organ (or by both replicates if only two samples were available), except when peaks were already derived from merged samples, as for the adult mouse organs. We finally resized the peaks to 1000 nt centered on the summit of the peak (or on the middle of the peak when the summit was not available).

**Definition of regulatory and inactive regions.** In each species, we defined as promoters the 1000 nt located upstream of a stable transcript. Putative enhancers in human and mouse were initially defined as DHSs overlapping an H3K27ac and/or an H3K4me1 peak. The resulting set of enhancers was further filtered to exclude loci located closer than 1000 nt from any H3K4me3 peak from any organs/tissues (including the extended dataset) or that overlap the 1000 nt region upstream of the TSS or the exons of any (stable or unstable) transcript. We further downloaded enhancer sets defined using CAGE from human and mouse[4] (permissive enhancers phase 1 and 2). These loci were subjected to the same filtering process described above, and then included in the final list of putative enhancers.

To define the set of inactive regions, we sampled from the human and mouse genome up to 1.5 million non-overlapping 1000 nt loci, and then removed from this list all loci mapping closer than 1000 nt from: (a) any DHS or any H3K4me1, H3K4me3 or H3K27ac peak from all organs from the core and extended dataset. Regions marked by active chromatin modifications but not DNase hypersensitive were excluded as they could potentially represent elements active in untested tissues or developmental stages; (b) any exon from all assembled (stable or unstable) or annotated transcripts[53,54] (GENCODE versions 27 for human and version vM16 for mouse); (c) any high identity (95%) segmental duplication or high identity (95%) repeat element annotated in the UCSC database[55].

**Definition and comparisons of P/E elements.** Coordinates of all regulatory and inactive regions from human and mouse were converted on the macaque and rat genome, respectively, using LiftOver[52] (with -minMatch = 0.6) to define their orthologous loci. A two-way liftOver conversion (species A—>species B—>species A) was adopted to avoid ambiguous orthology definitions that may result from genomic duplication events. We defined as P/E elements all human and mouse enhancers whose orthologous loci in macaque and rat, respectively, overlapped the 500 nt upstream of the TSS of a stable transcript. To avoid the inclusion of false positives due to potential 5′-truncated isoforms, we excluded from our final P/E elements dataset those associated to shorter isoforms of transcribed loci with an alternative, upstream promoter in macaque or rat. As a control, we identified all inactive regions in human and mouse whose orthologous sites in their sister species

overlapped the 500 nt upstream of the TSS of a stable transcript, and then compared the frequency of these loci with the frequency of P/E elements in each core sample with a Fisher's exact test. P/E elements were analysed separately to assess whether the associated transcript in macaque or rat corresponded to a new isoform of a transcribed locus present in the sister species (extended P/E element) or to a newly transcribed locus (novel P/E element). Shared transcribed loci between orthologous species were defined by the overlap of the orthologous sequences of macaque/rat transcripts (determined using liftOver) with reconstructed or GENCODE annotated transcripts in human/mouse.

We observed a significant difference in GC-content between the sets of enhancers and inactive regions (with an ortholog in the sister species). To control for the GC-content effect, we resampled sets of enhancers and inactive regions with a similar GC distribution. To this aim, we considered only enhancers with a GC-content lower than 46% (for human) or 41% (for mouse). These thresholds, roughly corresponding to the mode of the GC distribution of the enhancers set in the two species, were chosen as they represented the maximum value below which the GC distribution of inactive regions overlapped completely the GC distribution of the enhancers. Then, we subsampled up to 30,000 enhancers from each species (in order to have similar numbers of regions tested in both lineages), and for each locus we selected an inactive region with the most similar GC content. With this approach, we obtained sets of enhancers and inactive regions with statistically similar GC distributions and used these data to compare the frequency of P/E enhancers to that of inactive regions orthologous to promoters.

**ChIP-seq analysis of P/E elements.** To further support the promoter functionality of P/E promoters in rat and macaque, we compared their H3K4me3 profiles to those of other regulatory elements using liver ChIP-seq data obtained from Villar et al.[26] and aligned on the respective genomes using bwa aln[56]. In both species, the enhancer set corresponded to the orthologous putative enhancers projected (using liftOver) from their sister species. The promoters corresponded to putative conserved promoters, i.e. to macaque/rat promoters orthologous to human/mouse promoters, respectively (based on liftOver conversion). Only enhancers and promoter active in liver were used. Finally, we compared the average H3K4me3 coverage, normalized using the average input coverage, between all classes of regulatory elements with a Mann–Whitney $U$-test.

**PhastCons analysis.** We extracted the phastCons scores from P/E enhancers and other enhancers in human and mouse and plotted them over the whole regulatory region (pseudoscaled to 1000 nt) and over the neighboring 1000 nt regions using SeqPlots[57]. Heatmap clusters were defined using k-mean clustering. The phastCons data used corresponded to the "hg38.phastCons20way" for human (multiple alignments of 17 primates and three mammals) and the "mm10.60way.phast-Cons60wayGlire" for mouse (multiple alignment of 8 glires).

**Polarization of turnover events.** To define the directionality of the turnover events, we evaluated the presence of putative promoters or enhancers in regions syntenic to P/E elements active in liver in marmoset and rabbit. In marmoset and rabbit, enhancers corresponded to H3K27ac peaks not overlapping any H3K4me3 peak, the 1000 nt upstream of any (stable or unstable) assembled transcript, or any exon of these transcripts; promoters were defined as the 1000 nt upstream of the TSS of a stable transcript. Macaque and rat liver P/E element coordinates were converted in their outgroup species genome using LiftOver (with -minMatch = 0.4), and we then evaluated the overlap between the converted coordinates and the annotated regulatory regions.

To estimate the rate of regulatory element loss, we identified the orthologous loci of liver promoters and enhancers from human and mouse in the other species from the same clade (macaque/marmoset and rat/rabbit, respectively), and kept only those loci that aligned to all species of the same clade. We then identified human/mouse promoters associated (i.e. overlapping the 500 nt upstream of the TSS) to a stable liver transcript in marmoset/rabbit (ancestral promoters). Similarly, we defined as ancestral enhancers the human/mouse enhancers which overlapped an H3K27ac peak and that were not associated to a promoter or an H3K4me3 peak in marmoset/rabbit. The same approach was used to define human/mouse promoters and enhancers conserved in macaque/rat. Ancestral promoters and enhancers, which were not conserved in macaque/rat were defined as lost.

To estimate the fraction of ancestral promoters corresponding to P/E elements, we further calculated the overlap of primate and rodent P/E elements coordinates converted in marmoset and rabbit, respectively, with the putative promoter of any stable transcript (from any organ).

**Sequence composition of regulatory elements.** We extracted and compared the GC content (using the nuc tool from BEDTools[57]) and CpG dinucleotides frequency of different classes of regulatory elements in human and mouse using a Mann–Whitney $U$-test. The same features were compared between orthologous P/E elements in both lineages with a Wilcoxon signed-rank test. Finally, we estimated whether the magnitude of the evolutionary change in GC content or CpG frequency at P/E regions was higher than that measured at inactive regions (defined above) to control for global skews in sequence composition. This was done by

resampling 10,000 times from the set of inactive regions the same number of P/E elements, and then comparing the mean GC and CpG difference in P/E elements with the distribution of differences from the resampled set.

**Core promoter motifs on regulatory elements.** Core promoter motifs (POLII collection) were obtained from the JASPAR 2018 database[58]. We compared the relative enrichment of core promoter motifs between P/E elements and other promoters and enhancers using AME[58] with parameters --method ranksum and --scoring max to evaluate differences in the maximum strength of the motifs.

**U1/PAS motif composition of regulatory elements.** We defined the U1 nucleotide motif with the de novo tool from HOMER[59] (version 4.8.3) using as input the FASTA sequences of the 50 bp around each splicing donor site (−25/+25 nt) from 50,000 randomly selected human GENCODE transcripts. A similar approach was adopted to define the PAS motif using a set of experimentally defined PASs from human[60]. We determined the genome-wide location of U1 and PAS sites with the scanMotifGenomeWide tool from HOMER[59]. To compare the distribution of the U1 and PAS motifs around P/E elements, we considered the most upstream TSS of all P/E-associated transcripts in macaque and rat and projected their location in the corresponding sister species using BLAT[61]. We considered only novel P/E elements, given that U1/PAS motifs from downstream transcripts in extended P/E loci might have conflated the U1/PAS signal in mouse and human. We compared the density of U1 and PAS motifs over 1 kb up- and downstream of the P/E-associated TSS in each species using a Wilcoxon signed-rank test. The same approach was used to compare the density of these motifs between sister species. The proximity of the closest U1 or PAS site up- or downstream of the P/E-associated TSSs was calculated using closestBed from BEDTools[57]. To compare the distribution of U1/PAS sites at enhancer elements, we considered only CAGE-defined enhancers in human and mouse, and used their annotated CAGE peak as the TSS of each element. We used liftOver to define the orthologous loci of human and mouse enhancers in macaque and rat, respectively. We then averaged the number of U1 and PAS motifs located 1000 bp up- and downstream of each enhancer TSS and compared their density to that of P/E elements in all species.

**Code availability.** All processing of genomic coordinates was performed using tools from BEDTools suite[57] (version 2.25.0), samtools[56] (version 0.1.19) and in-house scripts. All statistical analysis was performed in R[62] using two-tailed tests (except when otherwise stated). The code used to perform all analyses presented in this paper is available as Supplementary Data 9.

## Data availability

Raw and processed datasets from this study have been submitted to the NCBI Gene Expression Omnibus under the accession [GSE114191].

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

## Acknowledgements

We thank K. Harshman and the Lausanne Genomics Technology Facility for high-throughput sequencing support; I. Xenarios and the Vital-IT computational facility for computational support; the Kaessmann group and Ana Claudia Marques for helpful discussions. The overall research project was supported by a grant from the European Research Council (Grant: 615253, OntoTransEvol) to H.K. F.N.C. was supported by a SNSF Early Postdoc.Mobility fellowship (P2LAP3_171808).

## Author contributions

F.N.C. and H.K. designed the study. F.N.C. conducted the analysis. A.L. and J.H. performed the experiments. F.N.C, M.W. and H.K. wrote the manuscript.

## Additional information

**Competing interests:** The authors declare no competing interests.

