## [Peer Review File · Nature Communications]

Reviewers' comments:

Reviewer #1 (Remarks to the Author):

The manuscript by Carelli and colleagues describes an investigation of the existence of evolutionary repurposing of regulatory elements from enhancers to promoters. The authors test a hypothesis that was formed by the Sharp lab a few years back, suggesting that changes in the U1/PAS axis can lead to the stabilization of transcripts. Supported by several reports on the similarities between enhancers and promoters and that enhancers in many cases also seem to have their own core promoters, Carelli and colleagues investigate shifts between regulatory functions in orthologous regions and how these relate to changes in the U1/PAS axis. The study is important to the field and will likely be of wide interest. Below, I raise some concerns and suggestions for improvements.

1) The major concern I have with the study in its current form is the implied assumption of enhancer function from ChIP-seq marks. First of all, such predicted enhancers have quite low in vitro validation rates suggesting that many predicted enhancers in this study are false positives. Secondly, even though the authors mention the problems with H3K4me3 vs H3K4me1 in determining regulatory function both in the introduction and in the discussion, they still rely on an enhancer definition from DHS + H3K27ac or H3K4me1 and an absence of H3K4me3. Apart from low in vitro reporter validation, a potential problem with this approach is that, since the level of H3K4me1 and H3K4me3 reflect transcriptional activity (e.g. Core et al, 2014), the authors might focus on an evolutionary repurposing of transcriptionally inactive promoters to transcriptionally active promoters rather than enhancers->promoters. This evolutionary change is in itself interesting but it is important that the authors clearly state their assumptions and discuss the limitations of their approach. In light of the relatively small number of repurposing events found, there is a risk that a considerable fraction of these are false positives.

2) The chromatin marks (and regulatory function) of promoters can differ between tissues of the same species (e.g. DOI 10.1038/nature14217). There is therefore a risk that some extended P/E:s are in fact P/P:s if more tissues were considered. The authors should comment on the potential limitations when only considering a handful of tissues. Related to this, from the methods section it is unclear if predicted enhancers from ChIP-seq data were filtered to be distal to annotated gene TSSs.

3) How do core promoter elements differ between P/E enhancers and other promoters/enhancers? The authors identify minor (but significant) differences in GC content and CpG frequency between P/E enhancers and other enhancers and suggest that this might reflect the ability to repurpose into promoters. Given the existence of core promoter elements also in enhancers, albeit often weaker, a naturally question is whether the P/E enhancers have stronger core promoters than other enhancers or if P/E promoters have specific core promoter elements.

4) How do P/E promoters compare to other (conserved) promoters in their ChIP-seq coverage? Figure 1D would benefit from having conserved promoters included. Are P/E promoters weaker?

5) Supplementary figure 4 and associated main text would benefit from including also the number of conserved promoters and enhancers (i.e. P-to-P and E-to-E) in order to put the number of repurposing events into perspective. In addition, the number of inactive-to-E and inactive-to-P should, if possible, also be compared in order to better judge the frequency of this evolutionary event.

Reviewer #2 (Remarks to the Author):

Authors study the turnover in the roles played by promoters (P) and enhancer (E) through time in mammals. They acknowledge that insertions and deletions, including transposable element insertions, play a major role in the evolution of regulatory regions but since many regions have been observed not to hold the same functionality between species and enhancers and promoters have overlapping features they explore if there are transitions from P to E or E to P in primates and rodents (i.e., two comparisons where there is or they have produced enough sequence and functional information to perform this comparisons). They identified 449 regions with turnover. For a few they infer ancestral state and conclude that E to P repurposing occurs more often than P to E and the features that facilitate this are the GC content and the evolution of splicing donor sites. The evaluation of the impact of this phenomenon is very interesting and it is relevant to the evolution of regulatory regions in new genes and new exons. They also conclude that de novo evolution of regulatory regions is not common. I have some comments and I also have some questions because I would like to understand exactly how the authors interpret their observations.

Comments:

1. Authors infer the ancestral state and conclude that E to P repurposing occurs more often than P to E. There is also a discussion about time/divergence eliminating the signatures of the functional modules in the outgroup. I think that, if divergence eliminates the signatures of the functional enhancers faster than those of functional promoters, i.e., because they are more complex, they would observe the same pattern.
2. I find one aspect of the work unclear. Authors mention that the studied regions are putative functional enhancers/promoters but they also state that the “detected events are not necessarily selectively preserved or of phenotypic relevance” (Page 4). Are the authors saying that these regions can be regions that mimic enhancers/promoters without being used as actual enhancers/promoters? What is the likelihood of that?
3. Related to question 2, I wonder how do they envision an enhancer can be lost to give rise to a promoter and what can be the functional consequences of this. Does this mean that the regulation (tissue expression) of a nearby gene changed? Is there any evidence of a change in expression pattern close to where the new promoter evolves? Is there any easy way to check this?
4. Related to question 2, if the regions just mimic enhancers/promoters without being actual enhancers/promoters, could it be that these regions mimicking a E or P evolved de novo?

Reviewer #3 (Remarks to the Author):

In the manuscript entitled “Functional repurposing of regulatory element activity during mammalian evolution” by Carelli et al., the authors use a comparative functional genomics approach to provide evidence for the functional repurposing of distal enhancer elements into novel promoters during mammalian evolution. Their approach was done simultaneously for both the primate and rodent lineages, seeing similar evolutionary trends for each clade. They identified several hundred enhancer elements from human and mouse that have been repurposed into promoters – either of novel genes or as upstream alternative transcription start sites – in macaque or rat, respectively. By looking at a third outgroup species for each clade, they were able to identify the directionality of these enhancer/promoter switches in many cases, finding that the majority of them seem to be an ancestral enhancer that has been repurposed into a promoter. Furthermore, the authors find small, but significant, changes in DNA sequence composition associated with these repurposed enhancers, including increased GC content and CGI density, as well as changes in the U1 and PAS density that the authors argue may lead to increased transcriptional stability.

Overall, the conclusions from the authors are convincing and supported by the data and analyses presented in the manuscript. Their particular approach is novel and well-thought out. One limitation of the work is the limited number of inter-species comparisons they were able to make due to the limited number of datasets collected, with an understandably heavy focus on human and mouse. They did not look at the chromatin signatures in multiple tissues in the other lineages, and so were unable to do the reciprocal comparisons. Another limitation: they could have used more appropriately matched controls than random genomic sequences. Because of the relatively low number of unambiguous examples (ca. 200 P/E elements out of >100,000 total elements from L151 and L157) the authors explore, there are a number of overly strong conclusions and/or statements that should be toned down, see editorial comments.

Major comments

1. The GEO accession numbers are listed as “pending” in the manuscript. The data should have been submitted to a database prior to submission of the manuscript, and made available to reviewers through a private link – which is possible in GEO, ArrayExpress and most other databases. Generally, I refuse to even review papers without access numbers that can be cross-checked. Any further review request I receive from Nature Communications will be more carefully scrutinized for data transparency before my acceptance.
2. The authors appear to select the best example either from the primate or the rodent lineage to showcase in the main manuscript, while hiding the less-convincing examples in the supplementary figures (Sup. Figures 1,5-10). To present a more unbiased view, the authors should present the results from both primate and rodent clades, side-by-side, in the main figures.
3. The control set of random regions used in a number of analyses are often not best suited for purpose. First, the authors provide the number control regions in the method section before filtering, but not the number that is left after filtering through which it would be easier to estimate how much of the genome is in their control set (L448-452). Second, in the result section (L186-187) the authors use this control set to conclude that de novo origination of regulatory activity is rare. Considering that the control is just a subset of the genome, it is not surprising that they find little overlap with newborn regulatory elements. A better control would be using the whole regulatory non-active section of the genome, for example by approximating it as the non-DNAse hypersensitive regions. Furthermore, random background genome, as in L258, is unsuitable as a background set for analysis of what sequence features are enriched within P/E elements; intuitively, this background should be other enhancer elements that are not P/E.
4. Some figure panels are poorly presented and explained. Before publication, the following ought to be rectified. Figure 1 panel d labeling should be more carefully considered. ChIP-seq read enrichment in Figure 1d and Supplementary Figure 1 could also include orthologous promoters, not only enhancers, to give a better insight into the meaning of the read enrichment changes of repurposed elements. Also in Figure 1d, was this analysis done on all P/E elements, or only for those active in liver? In the boxplots in Figure 3a-d and Supplementary Figure 5 only the P/E elements boxes have notches, the other three regulatory types do not. Why is that the case? In Panel 3e, it is impossible to understand what was done. Not explained well in main text, figure legend, or within the images. Panel 3f uses inappropriate background for their analysis (see major comment above). Sup. Figure 2/3 – The authors should choose a random set of unmapped regions with matched GC content for control above background analyses. Figure 4, Sup Figures 6-10 – For the U1 and PAS analysis, it would be informative to include the human/mouse enhancers that do not get repurposed into the analysis, so to evaluate whether their U1/PAS distribution differs from the P/E elements.
5. The authors use the highly outdated Ensembl database release 73, which was released in September of 2013 and is no longer even publicly available. The most current release of Ensembl is 91 (!), and contains a significant improvement to the macaque genome assembly originally made available in release 86. If they must use such outdated resources, the authors should include a supplementary figure that confirms that the number of total promoters, enhancers and P/E elements would not change if they used a more current versions of genome assemblies. On

the positive side, reanalysis using better underlying assemblies could improve the significance of differences in the GC content in Supplementary Figures 5b and 5d.

Editorial & other comments

1. Abstract. Because the number of examples is actually very small (low hundreds), many conclusions should be softened, sometimes substantially. For instance, I would strike the entire second to last sentence of the abstract (Overall, ... element functions.).
2. Please compress the introduction in length by about 35%; too wordy.
3. L99-100. This sentence is extremely awkward. In general, the use of "quotations" or (parentheses) should be studiously avoided wherever possible in formal scientific English.
4. All examples of parentheses should be inspected for removal, I will not list them all, as they are very many.
5. L94-95 "Interestingly, recent work reported the frequent evolutionary emergence and decay of promoters and enhancers in mammals." sounds as if promoters and enhancers turnover at equal frequencies, though Villar and mouseENCODE have shown that enhancers turnover more rapidly – as the authors themselves describe later in the manuscript's results section.
6. The discussion on the directionality of regulatory changes on pages 7-8 is based on data from a limited number of tissues in the outgroups (macaque and rabbit). Promoters are annotated through transcriptional data produced by the authors in 4 tissues, while H3K27ac and H3K4me3 data from Villar from only one tissue is used to annotate enhancers. Considering that a significant number of tissue-specific promoters and enhancers are missing in these datasets, I do not feel that the authors have enough evidence to comment on final ratios and directionality of repurposing events – they can only see a general trend on a subset of tissues. Their discussion on this might need to be toned down.
7. Page 17 L526 should "leftmost TSS" instead be most downstream/upstream in relation to gene? "left" does not seem to have an obvious meaning in relation to genomic coordinates.
8. L185-187. It may be prudent for the authors to note that their described functional repurposing is also quite rare. Remove quotes around "de novo"
9. In L194, should the source data be cited? (optional)
10. L202-218 has so many parentheses that I could hardly follow what they were (saying).
11. L278. 'Regulators' is the wrong word here.
12. L279. Strike "it has been shown that"
13. L286. Strike "In these species"
14. L292-294 Painfully awkward conclusion sentences.
15. L318-319. Please reword to "U1/PAS axis may contribute to...". Also, Sharp paper's hypothesis specified genes, not loci.
16. L325. Strike "strong". Conclusion is overstated: 200 examples out of 100+ thousand.
17. L330-349 need heavy editing. Bad paragraph copy.
18. L350-367. In contrast, a very nice paragraph.
19. L394-397. A bit too obsequious towards Sharp, tone down.

Reviewer summary

This manuscript makes a significant contribution to the field of evolutionarily genomics and sheds light on an interesting phenomenon likely to be important in driving mammalian evolution.

We would like to thank the reviewers for appreciating our work and the very constructive and useful comments, which helped us to improve the manuscript.

Reviewer #1 (Remarks to the Author):

The manuscript by Carelli and colleagues describes an investigation of the existence of evolutionary repurposing of regulatory elements from enhancers to promoters. The authors test a hypothesis that was formed by the Sharp lab a few years back, suggesting that changes in the U1/PAS axis can lead to the stabilization of transcripts. Supported by several reports on the similarities between enhancers and promoters and that enhancers in many cases also seem to have their own core promoters, Carelli and colleagues investigate shifts between regulatory functions in orthologous regions and how these relate to changes in the U1/PAS axis. The study is important to the field and will likely be of wide interest. Below, I raise some concerns and suggestions for improvements.

We thank the reviewer for the appreciation of our work and her/his constructive comments, which we address in detail below.

1) The major concern I have with the study in its current form is the implied assumption of enhancer function from ChIP-seq marks. First of all, such predicted enhancers have quite low in vitro validation rates suggesting that many predicted enhancers in this study are false positives. Secondly, even though the authors mention the problems with H3K4me3 vs H3K4me1 in determining regulatory function both in the introduction and in the discussion, they still rely on an enhancer definition from DHS + H3K27ac or H3K4me1 and an absence of H3K4me3. Apart from low in vitro reporter validation, a potential problem with this approach is that, since the level of H3K4me1 and H3K4me3 reflect transcriptional activity (e.g. Core et al, 2014), the authors might focus on an evolutionary repurposing of transcriptionally inactive promoters to transcriptionally active promoters rather than enhancers->promoters. This evolutionary change is in itself interesting but it is important that the authors clearly state their assumptions and discuss the limitations of their approach. In light of the relatively small number of repurposing events found, there is a risk that a considerable fraction of these are false positives.

We agree with the reviewer that we cannot be sure in all instances that the set of human/mouse elements correspond to enhancers, which is why we had referred to the human/mouse sequences as "putative enhancers" in the beginning of the results section and several other places in the original manuscript version. In addition, we would like to emphasize the following points regarding the reviewer's comments:

A) We believe it is unlikely that many of the human/mouse sequences correspond to promoters (inactive in the four main studied tissues) or bivalent elements, because our enhancer annotation did not only rely on DHS data and histone modifications signatures typical of enhancers; it also required the absence of evidence of stable transcription, as assessed by GENCODE, our RNA-seq data, and H3K4me3 signals in a broad set of tissues, cell types and developmental stages (127 datasets in human, 238 in mouse) (lines 125-133 main text, lines 451-457 Methods, Supplementary Table 8).

B) It is also unlikely that many of the human/mouse sequences correspond to typical non-regulatory genomic sequences, given that our analyses revealed that these sequences are much more likely to correspond to promoters in the sister species compared to our control sequences (i.e., genomic sequences with no signature of regulatory activity), which together with other evidence suggests that the presence of regulatory activity facilitated regulatory innovation (lines 164-184).

C) A sizeable part (34% in human and 15% in mouse) of the putative enhancers analysed in our work was defined using CAGE data, which are known to have a significantly higher validation rate in vitro compared to chromatin modification-defined enhancers (Andersson et al. *Nature* 2014). It is therefore notable that when we only consider CAGE-defined enhancers in our dataset (see new analyses described in lines 365-369), we still observe significantly higher proportions of repurposing events (0.18% in primates, 0.28% in rodents) compared to our non-regulatory control set and that these proportions are very similar to those found in the whole dataset (0.18% in primates, 0.24% in rodents; lines 171-176). These observations suggest that most detected cases indeed represent bona fide repurposing events and that the "de novo" emergence of regulatory sequences is comparatively rarer.

D) In this context, it is also noteworthy that recent work suggests that in vitro validation essays performed on episomal constructs may underestimate the proportion of true enhancers among those defined by chromatin signatures (Muerdter et al. *Nat. Methods* 2018), which seems higher when the same constructs are integrated into the genome (Inoue et al. *Genome Res.* 2017).

Overall, the aforementioned observations support the notion of at least large proportions of bona fide repurposing events in our data, although, as acknowledged above we agree with the reviewer that we cannot be sure for all cases and therefore generally speak of "putative enhancers". In addition, we added a new paragraph in the Discussion section (lines 356-369), which explicitly discusses the uncertainty in annotating enhancers, but also the evidence supporting actual regulatory repurposing, as outlined above.

2) The chromatin marks (and regulatory function) of promoters can differ between tissues of the same species (e.g. DOI 10.1038/nature14217). There is therefore a risk that some extended P/E:s are in fact P/P:s if more tissues were considered. The authors should comment on the potential limitations when only considering a handful of tissues.

As also pointed out above (reply A, reviewer comment 1), we actually had rather rigorously filtered out all regions with (putative) promoter activity (i.e., H3K4me3 enrichment and/or association to reconstructed or annotated transcripts) in samples representing a large number of adult and embryonic tissues (127 in human, 238 in mouse). We therefore believe it is unlikely that many of the human/mouse sequences correspond to promoters or bivalent elements. We now optimized the relevant main manuscript text to better clarify our approach and now also state that "...nonetheless, we cannot exclude that some putative enhancers might have promoter activity in other tissues or developmental stages." (lines 133-134).

Related to this, from the methods section it is unclear if predicted enhancers from ChIP-seq data were filtered to be distal to annotated gene TSSs.

Yes, predicted enhancers were filtered to be at least 1,000 bp upstream of any TSS and 1,000 bp away from any H3K4me3 peak. They were also required to not overlap with any annotated or reconstructed exon. We now optimized the relevant sentence in the Methods section to make this clearer (lines 472-474).

3) How do core promoter elements differ between P/E enhancers and other promoters/enhancers? The authors identify minor (but significant) differences in GC content and CpG frequency between P/E enhancers and other enhancers and suggest that this might reflect the ability to repurpose into promoters. Given the existence of core promoter elements also in enhancers, albeit often weaker, a naturally question is whether the P/E enhancers have stronger core promoters than other enhancers or if P/E promoters have specific core promoter elements.

We thank the reviewer for the interesting idea, which is however hard to put into practice, given that it is hard to predict promoter strength from sequences alone; that a precise TSS annotation required for detailed motif analyses is currently difficult, especially in macaque and rat; and that motif predictions may depend on overall sequence compositional changes. We nevertheless carried out a comparative analysis of core promoter motifs.

Specifically, we compared the relative enrichment of 13 core promoter motifs downloaded from the JASPAR2018 collection (JASPAR Collection POLII) between the different classes of regulatory elements (CGI- and non-CGI promoters, enhancers, P/E elements) in all species. This analysis revealed differences in the type of core promoter motifs relatively enriched at distinct regulatory regions: P/E elements are depleted for GC-rich motifs compared to CGI promoters and enriched for the same motifs compared to other enhancers, whereas no difference was detected between P/E elements and non-CGI promoters. The latter result is not observed in rat, likely due to the lower number of annotated CpG islands. Indeed, CpG islands in rodents are more CpG-poor compared to those in primates and most software tools tend to call fewer islands in these species. We did not have the same issue in mouse, given that we could use a set of functionally defined CpG islands for this species. Overall, our analysis suggests that differences in core promoter motif compositions between the different types of elements are consistent with differences/changes in GC content, which renders a functional interpretation of the observations difficult. Given the aforementioned caveats and that the functional relevance is unclear, we only briefly mention the results of this new analysis in the main text (lines 245-248) and details in the new Supplementary Table 6.

4) How do P/E promoters compare to other (conserved) promoters in their ChIP-seq coverage? Figure 1D would benefit from having conserved promoters included. Are P/E promoters weaker?

This is a very good suggestion and so we now included the H3K4me3 coverage measured over promoters conserved in the sister species in Fig. 1d, which now includes both macaque and rat comparisons. In both species, conserved promoters show a higher H3K4me3 coverage compared to P/E elements. This indicates that P/E promoters are indeed weaker than other promoters, as also reflected in the expression level of the associated transcripts (now included in Supplementary Fig. 1). All results are reported and discussed in lines 152-162.

5) Supplementary figure 4 and associated main text would benefit from including also the number of conserved promoters and enhancers (i.e. P-to-P and E-to-E) in order to put the number of repurposing events into perspective. In addition, the number of inactive-to-E and inactive-to-P should, if possible, also be compared in order to better judge the frequency of this evolutionary event.

We agree with the reviewer and created a new Supplementary Table 5, which includes:

- the total number of human/mouse enhancers and promoters
- the total number of human/mouse enhancers and promoters aligned in all species from each clade.
- the number of promoters and enhancers conserved in all species of the primate and glires clades (E-to-E and P-to-P)
- the number of enhancers and promoters shared by human/macaque and mouse/rat but absent in the outgroup species (inactive-to-E and inactive-to-P)

We now overall optimized the text pertaining to these specific analyses (lines 209-227), and to better highlight them, we now integrated the former Supplementary Fig. 4 in main Fig. 2 as panels b-d.

Reviewer #2 (Remarks to the Author):

Authors study the turnover in the roles played by promoters (P) and enhancer (E) through time in mammals. They acknowledge that insertions and deletions, including transposable element insertions, play a major role in the evolution of regulatory regions but since many regions have been observed not to hold the same functionality between species and enhancers and promoters have overlapping features they explore if there are transitions from P to E or E to P in primates and rodents (i.e., two comparisons where there is or they have produced enough sequence and functional information to perform this comparisons). They identified 449 regions with turnover. For a few they infer ancestral state and conclude that E to P repurposing occurs more often than P to E and the features that facilitate this are the GC content and the evolution of splicing donor sites. The evaluation of the impact of this phenomenon is very interesting and it is relevant to the evolution of regulatory regions in new genes and new exons. They also conclude that de novo evolution of regulatory regions is not common. I have some comments and I also have some questions because I would like to understand exactly how the authors interpret their observations.

We thank the reviewer for appreciating our study and her/his useful comments, which we address in detail below.

Comments:

1. Authors infer the ancestral state and conclude that E to P repurposing occurs more often than P to E. There is also a discussion about time/divergence eliminating the signatures of the functional modules in the outgroup. I think that, if divergence eliminates the signatures of the functional enhancers faster than those of functional promoters, i.e., because they are more complex, they would observe the same pattern.

To specifically address this potential bias, we had designed our control analysis in which we compare the rates of functional repurposing and loss of regulatory activity. This analysis reveals indeed that putative ancestral enhancers are lost at a higher rate than ancestral promoters, but that an even (much) higher proportion of functional turnover events involves ancestral enhancers compared to ancestral promoters. Therefore, the higher frequency of E-to-P vs P-to-E events observed cannot be explained by the higher evolutionary turnover of enhancer elements. Because of the reviewer's comment and importance of this analysis, we now further clarify its rationale and conclusions in the main text (lines 209-227) and added an explanatory graphical illustration in Fig. 2 (panels b-d).

2. I find one aspect of the work unclear. Authors mention that the studied regions are putative functional enhancers/promoters but they also state that the “detected events are not necessarily selectively preserved or of phenotypic relevance” (Page 4). Are the authors saying that these regions can be regions that mimic enhancers/promoters without being used as actual enhancers/promoters? What is the likelihood of that?

Thanks for pointing this out, so that we can clarify what we mean by this cautionary/prudent note. Especially in many mammalian (vertebrate) genomes, which have long intergenic sequences and the efficiency of selection is low (due to low effective population sizes), various sequence types, including elements with regulatory capacities, may spuriously arise during evolution but not have any impact on fitness; that is, they may, for example, drive transcription, but are not selectively preserved (i.e., they are functionally neutral) and have no impact on the phenotype. However, such "proto-promoters" (or proto-enhancers) may eventually be recruited as, for example, new gene promoters during evolution. Please refer to two reviews and a recent paper from our lab (Kaessmann et al. *Nat. Rev. Genet.* 2009, Kaessmann *Genome Res.* 2010, Carelli *Genome Res.* 2016) for interesting cases and a detailed discussion of proto-promoters and de novo promoter origins.

This means that at least some of the evolutionary changes in regulatory element activity detected in our study might not necessarily have any phenotypic implications. For example, in cases where enhancers that evolved into promoters with the capacity to give rise to stable transcripts, the promoter might represent a "proto-promoter" (instead of a new promoter associated with a new gene/transcript), in which case the newly transcribed locus is not under purifying selection. Notably, in any event, the detected events illustrate that functional repurposing facilitates regulatory innovation by providing raw material for regulatory evolution. We now optimized the overall discussion of the potential phenotypic relevance of functional repurposing (Discussion section: lines 413-425).

However, stimulated by the reviewer's comment, we performed PhastCons analyses with the aim to assess the presence/extent of purifying selection in P/E elements. This analysis reveals that the sets of P/E elements overall show signatures of purifying selection, which suggests that at least subsets of them acted as functional regulators (as enhancers and/or promoters) at least during some time during their evolutionary history in the studied primate/rodent lineages. Together with other observations, such as the CpG and U1 site shifts accompanying the repurposing events, the detected signatures of selection support the notion that at least a sizeable proportion of cases are of phenotypic relevance. The new results are shown in Supplementary Fig. 2 and discussed in Results section (lines 159-162) of the revised manuscript.

3. Related to question 2, I wonder how do they envision an enhancer can be lost to give rise to a promoter and what can be the functional consequences of this. Does this mean that the regulation (tissue expression) of a nearby gene changed? Is there any evidence of a change in expression pattern close to where the new promoter evolves? Is there any easy way to check this?

The detected enhancer repurposing events (i.e., changes of chromatin profiles and gain of stable transcription) may frequently not have functional consequences on the genes regulated by this enhancer, for two reasons:

First, the repurposing of an enhancer element into a promoter per se might not necessarily lead to the loss of its activity as a distal regulatory element (see e.g. the cited reference: Dao et al. *Nat. Genet.* 2017). As we point out in the introduction, there are regulatory elements (bivalent elements) exerting both promoter and enhancer functions. While we tried to avoid the inclusion of bivalent elements in our human and mouse enhancer set by filtering out regions with promoter signatures in a large set of adult and developmental samples (lines 125-133), we cannot assess whether the P/E elements in macaque and rat – in which we observe a promoter activity – retained (ancestral) enhancer capacities, due to a lack of suitable data.

Second, the regulatory activity exerted by enhancer elements is often carried out in concert with other distal elements, probably to ensure evolutionary robustness in gene expression. Two recent papers (Berthelot et al. *Nat. Ecol. Evol.* 2018; Danko et al. *Nat. Ecol. Evol.* 2018) indeed showed that mammalian genes characterized by numerous enhancers have more conserved expression levels than genes with simpler regulatory landscapes. The presence of multiple enhancers likely ensures the evolutionary stability of gene expression in the event of loss of one or a part of the ancestral distal regulatory regions.

Overall, the observed enhancer repurposing events may therefore often leave ancestral target gene expression unaltered. However, with currently available data, we cannot assess whether there is any impact, in particular because we would need to know the genes regulated by the respective enhancer, which would require dedicated data (e.g. 3C or 4C data for the studied tissues), given that the gene closest to the enhancer is often not the actual regulatory target (e.g., Sanyal et al. *Nature* 2012). We now

added the aforementioned aspects in the new discussion to clarify the point raised by the referee (lines 413-425).

4. Related to question 2, if the regions just mimic enhancers/promoters without being actual enhancers/promoters, could it be that these regions mimicking a E or P evolved de novo?

Our analyses demonstrate that enhancer signatures in human/mouse substantially increase the chance of observing promoter activity in macaque/rat compared to the genomic, non-regulatory background, which suggests that P/E elements (under purifying selection or not — see reply to reviewer comment 2) are unlikely to have (independently) emerged de novo in the primate rodent species/lineages studied; that is, annotated P/E elements had regulatory precursor elements in common human-macaque or mouse-rat ancestors, respectively (lines 164-184). However, the precursor elements in these ancestors may have emerged de novo earlier in evolution.

Reviewer #3 (Remarks to the Author):

In the manuscript entitled “Functional repurposing of regulatory element activity during mammalian evolution” by Carelli et al., the authors use a comparative functional genomics approach to provide evidence for the functional repurposing of distal enhancer elements into novel promoters during mammalian evolution. Their approach was done simultaneously for both the primate and rodent lineages, seeing similar evolutionary trends for each clade. They identified several hundred enhancer elements from human and mouse that have been repurposed into promoters – either of novel genes or as upstream alternative transcription start sites – in macaque or rat, respectively. By looking at a third outgroup species for each clade, they were able to identify the directionality of these enhancer/promoter switches in many cases, finding that the majority of them seem to be an ancestral enhancer that has been repurposed into a promoter. Furthermore, the authors find small, but significant, changes in DNA sequence composition associated with these repurposed enhancers, including increased GC content and CGI density, as well as changes in the U1 and PAS density that the authors argue may lead to increased transcriptional stability.

Overall, the conclusions from the authors are convincing and supported by the data and analyses presented in the manuscript. Their particular approach is novel and well-thought out. One limitation of the work is the limited number of inter-species comparisons they were able to make due to the limited number of datasets collected, with an understandably heavy focus on human and mouse. They did not look at the chromatin signatures in multiple tissues in the other lineages, and so were unable to do the reciprocal comparisons. Another limitation: they could have used more appropriately matched controls than random genomic sequences. Because of the relatively low number of unambiguous examples (ca. 200 P/E elements out of >100,000 total elements from L151 and L157) the authors explore, there are a number of overly strong conclusions and/or statements that should be toned down, see editorial comments.

We are grateful for the overall appreciation of our work by the reviewer and her/his constructive and helpful comments, to which we respond in detail below.

Major comments

1. The GEO accession numbers are listed as “pending” in the manuscript. The data should have been submitted to a database prior to submission of the manuscript, and made available to reviewers through a private link – which is possible in GEO, ArrayExpress and most other databases. Generally, I refuse to even review papers without access numbers that can be cross-checked. Any further review request I receive from Nature Communications will be more carefully scrutinized for data transparency before my acceptance.

We sincerely apologize for not having included a GEO reference to our newly produced RNA-seq data in our first submission — we were eager to submit the manuscript after data submission and overlooked the possibility of providing a private reviewer link. All data (both raw and processed) are now available under the accession number GSE114191 (line 602).

2. The authors appear to select the best example either from the primate or the rodent lineage to showcase in the main manuscript, while hiding the less-convincing examples in the supplementary figures (Sup. Figures 1,5-10). To present a more unbiased view, the authors should present the results from both primate and rodent clades, side-by-side, in the main figures.

The reasons for not always showing all results for both species sets were that we tried to avoid overloading the main figures (i.e., to keep them at a reasonable size) and that we preferred to display results for the clade more suitable/powerful for a given analysis (i.e. typically rodents, e.g. for the U1/PAS analysis), which we also explained in the main text. We also discussed all results in the main text and referring to the supplemental figures. However, motivated by the reviewer's comment, we now juxtapose rodent and primate results in figures in three additional cases, to allow for more comparisons directly from the main figures.

Specifically: (i) Figure 1d now includes the comparison of H3K4me3 levels between P/E elements, enhancers and promoters both for macaque and rat; (ii) Figures 3a-d now show the comparison of GC and CpG content values between P/E elements, enhancers and promoters in human and mouse, to highlight how P/E elements with enhancer signatures are significantly different from other enhancers in species from both clades (the corresponding results for macaque and rat is now included in Supplementary Figure 5), and (iii) Figure 4d now also includes the U1 density distribution plot for primates.

3. The control set of random regions used in a number of analyses are often not best suited for purpose. First, the authors provide the number control regions in the method section before filtering, but not the number that is left after filtering through which it would be easier to estimate how much of the genome is in their control set (L448-452). Second, in the result section (L186-187) the authors use this control set to conclude that de novo origination of regulatory activity is rare. Considering that the control is just a subset of the genome, it is not surprising that they find little overlap with newborn regulatory elements. A better control would be using the whole regulatory non-active section of the genome, for example by approximating it as the non-DNAse hypersensitive regions. Furthermore, random background genome, as in L258, is unsuitable as a background set for analysis of what sequence features are enriched within P/E elements; intuitively, this background should be other enhancer elements that are not P/E.

Apparently our description of the control regions used in our analyses was not sufficiently clear, leading to a misunderstanding — we apologize for this.

We had defined the controls exactly as suggested by the reviewer. That is, control regions in our analyses correspond to regions showing no signature of regulatory activity, as assessed by the large sets of DHS and histone modification data described in the paper and no overlap with exonic sequences. We also excluded recent segmental duplications and repeats (e.g., transposable elements), which would confound analyses. The length of each control sequence (1000 nt) was matched to the length defined for regulatory element annotation. In total, we thus annotated 181,688 control sequences in humans (corresponding to $\approx 7\%$ of the sequenced genome) and 83,288 sequences ($\approx 3.6\%$ of the genome) in mouse. We now optimized the relevant parts in the main text and Methods to better explain our definition of controls (lines 169-171, lines 342-342, lines 496-498, lines 507-509).

Our controls should therefore allow for reasonable (and currently best possible) estimations of the proportion of de novo origination of regulatory elements compared to the proportion of regulatory repurposing (lines 164-184 in the revised manuscript). The observed numbers should also provide an approximate idea of the absolute number of de novo events in the genome, given that the regulatory inactive portion of the genome defined in our work represents relatively large portions of the non-repetitive sequenced genomes of human and mouse.

Our reason to use the inactive sequences as a control in the analysis reported in the section “Sequence compositional changes probably contributed to repurposing” is to have a set of regions putatively not under selection, which would allow us to globally evaluate neutral changes in sequence composition that emerged during evolution between the studied pairs of species. We believe that enhancer elements are not really suitable for this purpose, given that they have regulatory functions that require specific sequences and that natural selection likely shapes the evolution of their sequences. Our set of enhancers has indeed a markedly higher GC content compared to the set of inactive regions, as shown in Supplementary Fig. 3 and as mentioned in the main text (lines 176-179). We therefore prefer to keep the analysis as is; that is, to use the large set of inactive regions as a control in the analyses pertaining to sequence compositional change.

4. Some figure panels are poorly presented and explained. Before publication, the following ought to be rectified. Figure 1 panel d labeling should be more carefully considered. ChIP-seq read enrichment in Figure 1d and Supplementary Figure 1 could also include orthologous promoters, not only enhancers, to give a better insight into the meaning of the read enrichment changes of repurposed elements. Also in Figure 1d, was this analysis done on all P/E elements, or only for those active in liver?

To address the reviewer's useful suggestions, we now include input-normalized H3K4me3 ChIP-seq coverage for conserved promoters in Figure 1d, as also pointed out in our reply to comment 4 of Reviewer 1. Notably, as also pointed out in our reply to comment 2 of this reviewer above, this figure panel now also includes H3K4me3 coverage distributions for macaque, which was previously in Supplemental Figure 1. The labelling in the figure has been changed to “extended P/E”, “novel P/E”, “orthologous enhancers”, “conserved promoters”, to make the content clearer. The conserved promoters correspond to the putative promoter regions of stable transcripts from rat and macaque that are orthologous to stable promoters from mouse and human, respectively. The analysis was performed considering P/E elements, other orthologous enhancers and conserved promoters active in liver. This is now explicitly mentioned in the Methods section (lines 518-527).

In the boxplots in Figure 3a-d and Supplementary Figure 5 only the P/E elements boxes have notches, the other three regulatory types do not. Why is that the case?

In the boxplots in Figure 3a-d, notches are indeed present for all distributions, but the very large sample sizes (i.e., number of promoters and enhancers) render the notches too narrow to be readily visible in several instances without magnification.

In Panel 3e, it is impossible to understand what was done. Not explained well in main text, figure legend, or within the images. Panel 3f uses inappropriate background for their analysis (see major comment above).

We thank the reviewer for pointing out that our explanation was not clear. We have now clarified our approach in the main text at lines 265-272, modified the labelling of Figure 3f, Supplementary Figures 6 and 7 and reworded the associated Figure legends. Please refer to our reply to comment 3 of this reviewer regarding the control/background used in the analysis underlying Figure 3f.

Sup. Figure 2/3 – The authors should choose a random set of unmapped regions with matched GC content for control above background analyses.

As also explained in our response to comment 3 of this reviewer, Supplementary Figure 3 (i.e., Supplementary Fig. 2 in the original manuscript) shows the overall distribution of GC content for the whole set of enhancers and inactive (control) regions. As suggested by the reviewer, the set of inactive regions used as controls for the enrichment analysis reported in Supplementary Figure 4 (i.e., former Supplementary Fig. 3) indeed correspond to random sub-samples with matched GC content (main text: lines 176-179, Methods: lines 506-516). We now also explain this better in the legend of Supplementary Figure 4, which was indeed not clear.

Figure 4, Sup Figures 6-10 – For the U1 and PAS analysis, it would be informative to include the human/mouse enhancers that do not get repurposed into the analysis, so to evaluate whether their U1/PAS distribution differs from the P/E elements.

We thank the reviewer for this useful suggestion. We have now included in the main text (lines 292-296) and in the new Supplementary Figure 8 the results of the analysis proposed by the reviewer. Briefly, given that our U1/PAS comparison relies on the identifications of motifs downstream of a given TSS, we focused our analysis on the set of CAGE-defined enhancers (not overlapping with P/E enhancers) characterized by the FANTOM consortium, which annotated the TSS associated to every enhancer peak. Given that these enhancers were defined through the detection of bidirectional transcription, we calculated the average number of U1/PAS motifs located within 1 kb on both sides of each element, and compared it to the number of U1/PAS motifs up- and downstream of each P/E element. The results of this analysis further highlight that the U1/PAS motif composition distinguishes P/E elements (regardless of their activity) from other classes of enhancers.

5. The authors use the highly outdated Ensembl database release 73, which was released in September of 2013 and is no longer even publicly available. The most current release of Ensembl is 91 (!), and contains a significant improvement to the macaque genome assembly originally made available in release 86. If they must use such outdated resources, the authors should include a supplementary figure that confirms that the number of total promoters, enhancers and P/E elements would not change if they used a more current versions of genome assemblies. On the positive side, reanalysis using better underlying assemblies could improve the significance of differences in the GC content in Supplementary Figures 5b and 5d.

We agree with the reviewer that our analyses should ideally be based on the latest genome assemblies/releases for all species in the study. We thus decided to rerun all analyses on the latest UCSC genome assembly/annotation versions for all species in our study (human: hg38; macaque: rheMac8; marmoset: calJac3; mouse: mm10; rat: rn6; rabbit: oryCun2). To further improve our analyses, we used the new and superior RNA-seq alignment tool STAR for the revision analyses, given that it outperforms TopHat2 (used in for the analyses in the previous version of the paper) in terms of both speed and accuracy. Finally, we note that we also used the latest versions (version 27 for human, version vM16 for mouse) of the GENCODE annotation available for both mouse and human for the new analyses.

The numbers of P/E elements in the new sets are very similar to the previous sets, with a total of 445 elements in the new set (184 in primates, 261 in rodents) compared to 449 in the original set (191 in primates, 258 in rodents). However, given the new (improved) genome assemblies, GENCODE annotations and alignment procedures, new P/E elements are detected while others are filtered out. That is, the new rodent set includes ≈68% of element of the original set and the primate set (affected by the much improved macaque genome version) retains ≈48% of the original set. However, we note that our filtering procedure is rather rigorous and most elements not retained in the new analyses likely still correspond to P/E elements but now do not (quite) pass our filtering criteria/thresholds anymore.

Thus, detailed inspection reveals that in many cases elements are still annotated as enhancers in human/mouse but now are just below the FPKM > 1 cutoff in macaque/rat, which would have qualified them as promoters (i.e., giving rise to stable transcription) in our analyses. Other previous P/E elements were not retained because they are now associated to shorter isoforms of transcribed loci with an alternative, upstream promoter. In our procedure, we conservatively exclude these regions, because they could potentially represent truncated versions of the longer transcript erroneously reconstructed by StringTie. The remaining previous non-retained P/E elements are now associated to a GENCODE transcript, and given that we conservatively exclude bivalent elements (with both enhancer and promoter signatures) in human/mouse in our analyses, these elements are now filtered out. Overall, we therefore likely underestimate the abundance of P/E elements for several reasons, which were partly already mentioned in the Discussion section (lines 344-350), but we prefer to remove potential false negatives (and minimize false positives) and thus retain a high confidence dataset for the analyses and as a future resource for the community.

In any event, it is important to note that the new data confirm the overall conclusions the original manuscript, with only subtle changes, such as the observation of a significant increase in U1 sites density downstream of macaque P/E elements compared to the human ones, which now makes the primate finding more similar to what is observed in rodents.

Editorial & other comments

1. Abstract. Because the number of examples is actually very small (low hundreds), many conclusions should be softened, sometimes substantially. For instance, I would strike the entire second to last sentence of the abstract (Overall, ... element functions.).

We removed this sentence from the abstract as suggested by the reviewer.

2. Please compress the introduction in length by about 35%; too wordy.

We have reduced the length of the introduction from 893 to 693 words (about 22%). We believe that the remaining text is required to provide sufficient background information for our study.

3. L99-100. This sentence is extremely awkward. In general, the use of “quotations” or (parentheses) should be studiously avoided wherever possible in formal scientific English.

We modified this sentence to make it clearer (lines 86-88). It now reads: “This raises the possibility that some regulatory elements might experience changes in their activity during evolution — a process we refer to as regulatory repurposing.”

4. All examples of parentheses should be inspected for removal, I will not list them all, as they are very many.

We have carefully inspected the text, as suggested by the reviewer, and considerably reduced the number of parentheses.

5. L94-95 “Interestingly, recent work reported the frequent evolutionary emergence and decay of promoters and enhancers in mammals.” sounds as if promoters and enhancers turnover at equal frequencies, though Villar and mouseENCODE have shown that enhancers turnover more rapidly – as the authors themselves describe later in the manuscript’s results section.

We now refined this sentence (lines 82-84), which now reads: “Interestingly, recent work reported the frequent evolutionary emergence and decay of enhancers²⁶ and, at a lower rate, promoters²⁷ in mammals.”

6. The discussion on the directionality of regulatory changes on pages 7-8 is based on data from a limited number of tissues in the outgroups (macaque and rabbit). Promoters are annotated through transcriptional data produced by the authors in 4 tissues, while H3K27ac and H3K4me3 data from Villar from only one tissue is used to annotate enhancers. Considering that a significant number of tissue-specific promoters and enhancers are missing in these datasets, I do not feel that the authors have enough evidence to comment on final ratios and directionality of repurposing events – they can only see a general trend on a subset of tissues. Their discussion on this might need to be toned down.

We agree with the reviewer that the low number of liver P/E elements investigated might limit the confidence of our results, and we therefore added the following new sentence at lines 224-227 to clarify this issue: “In any event, our primate analyses suggest that the higher rate of enhancer repurposing cannot be explained by the higher turnover rate of enhancers compared to promoters, although future work on additional closely related sets of species and organs will be needed to confirm this pattern.”

7. Page 17 L526 should “leftmost TSS” instead be most downstream/upstream in relation to gene? “left” does not seem to have an obvious meaning in relation to genomic coordinates.

We agree and now corrected this to: “the most upstream TSS” (line 580).

8. L185-187. It may be prudent for the authors to note that their described functional repurposing is also quite rare. Remove quotes around “de novo”

In light of our previous comments (see point 3), we now changed the sentence as follows (lines 179-184): “These data corroborate the hypothesis that P/E elements likely correspond to ancestral regulatory regions that experienced evolutionary changes in their regulatory activity in the last 25-29 millions of years. Ancestral regulatory capacities of genomic sequences therefore facilitated regulatory innovation in mammals, while our analyses also suggest that a sizeable number of lineage-specific regulatory elements may have emerged *de novo* from the large inactive portion of the genome.”

9. In L194, should the source data be cited? (optional)

Yes, it should. We therefore added the reference to Villar et al. (2015) at line 191.

10. L202-218 has so many parentheses that I could hardly follow what they were (saying).

We now have extensively modified and simplified the structure of the whole paragraph (lines 209-227).

11. L278. ‘Regulators’ is the wrong word here.

Thanks for pointing this out. We changed “regulators” to “elements” (line 280).

12. L279. Strike “it has been shown that”

We removed the words as suggested (line 281 of the revised manuscript).

13. L286. Strike “In these species”

Removed as suggested (line 288).

14. L292-294 Painfully awkward conclusion sentences.

We have now modified the sentence (lines 296-299) as follows: “In light of the higher enhancer-to-promoter repurposing rate (see above), these results suggest that the unique sequence and motif composition distinguishing P/E enhancers from other typical enhancers may predispose them to repurposing into novel promoters during evolution.”

15. L318-319. Please reword to “U1/PAS axis may contribute to...”. Also, Sharp paper’s hypothesis specified genes, not loci.

We reworded the sentence as suggested by the reviewer (lines 324-326). Furthermore, we note that although the paper from Wu and Sharp focuses on the role of U1/PAS axis changes in the emergence of new genes, it also describes the emergence of new stable transcripts from promoters or enhancers as the first step, and the acquisition of an ORF as the last step in this process. As we do not now at this stage to what extent the detected cases correspond to functional new genes rather than transcribed loci, we refer to both possibilities in the revised sentence.

16. L325. Strike “strong”. Conclusion is overstated: 200 examples out of 100+ thousand.

We removed the word “strong” as suggested (sentence lines 332-335).

17. L330-349 need heavy editing. Bad paragraph copy.

We have extensively reworded the paragraph (lines 337-354).

18. L350-367. In contrast, a very nice paragraph.

Thank you.

19. L394-397. A bit too obsequious towards Sharp, tone down.

The sentence now reads: “These transcripts may, potentially, represent new genes, which would support the hypothesis put forward by Wu and Sharp” (lines 422-423).

REVIEWERS' COMMENTS:

Reviewer #1 (Remarks to the Author):

The authors have adequately addressed all my previous comments and made the sufficient changes to the text and figures.

Reviewer #3 (Remarks to the Author):

This resubmission has addressed thoroughly all the major and minor comments from all reviewers. It is ready for publication, and will be an interesting and widely cited manuscript in the field.

The only remaining (optional) comment I have is that the title would be more accurate if it read "Repurposing of promoters and enhancers during mammalian evolution." That revised title sounds more narrow, but in fact the paper could be even more highly cited because of that specificity. Most people think of the activity of promoters and enhancers, and few think generally about regulatory elements, which sounds static.